# Efficacy of JAK1/2 inhibition in murine myeloproliferative neoplasms is not mediated by targeting oncogenic signaling

Sivahari Prasad Gorantla[1,2,13], Michael Rassner[1,3,13], Kirstyn Anne Crossley[1,4], Tony Andreas Müller[1,5], Teresa Poggio [1], Shifa Khaja Saleem[1], Helen Kleinfelder[1], Sudheer Madan Mohan Gambheer[1], Cornelia Endres[1], Sabina Schaberg[1], Dominik Schmidt [1], Gerin Prince [2], Irene Gonzalez-Menendez[6,7], Detlef Bentrop[8], Rainer Trittler[9], Svetlana Rylova[10], Dietmar Pfeifer[1], Geoffroy Andrieux [11,12], Leticia Quintanilla-Martinez[6,7], Anna Lena Illert[1,11], Nikolas von Bubnoff [2], Robert Zeiser[1] & Justus Duyster [1,11] ✉

Ruxolitinib is a potent JAK1/JAK2 inhibitor, approved for the treatment of primary myelofibrosis (PMF) patients based on the concept of inhibition of oncogenic signaling. However, the effect of ruxolitinib on JAK2-V617F allelic burden is modest, suggesting that inhibition of JAK2-V617F signaling-driven clone expansion is not the main mechanism of action. We evaluate whether ruxolitinib mainly blocks the proliferation of the malignant clone or exerts its effects also by targeting non-malignant cells. Therefore, we develop two JAK2-V617F-driven myeloproliferative neoplasm (MPN) mouse models harboring ruxolitinib resistance mutations. Mice carrying ruxolitinib-resistant JAK2-V617F-driven MPN respond to ruxolitinib treatment similar to mice with ruxolitinib-sensitive JAK2-V617F MPN with respect to reduction of spleen size, leukocyte count and pro-inflammatory cytokines in the serum. Ruxolitinib reduces pro-inflammatory cytokines in both stromal cells and non-malignant hematopoietic cells. Using a rigorous ruxolitinib resistance mutation approach, we can prove that ruxolitinib acts independent of oncogenic JAK2-V617F signaling and reduces the main features of MPN disease such as spleen size and leukocyte counts. Our findings characterize the mechanism of action for ruxolitinib in MPN.

Janus kinase 2 (JAK2) is a cytoplasmic tyrosine kinase that plays a major role in hematopoiesis and cytokine mediated signaling[1,2]. Myeloproliferative neoplasms (MPNs) encompass a spectrum of hematologic disorders, including polycythemia vera (PV), essential thrombocythemia (ET), and primary myelofibrosis (PMF). A hallmark of these conditions is the somatic activating mutation JAK2 V617F, which involves a substitution of valine to phenylalanine at codon 617 in the pseudokinase domain of the JAK2 gene. The prevalence of JAK2 V617F varies among MPN subtypes (90% of patients with PV, 50% with ET, and 50%

with PMF)[3–7]. Additionally, this mutation can be seen at lower frequency in other hematologic disorders, for instance chronic myelomonocytic leukemia, mylodysplastic neoplasms, or systemic mastocytosis[4,8]. JAK-STAT signaling pathways are important drivers of pro-inflammatory responses that are characteristic for MPNs, such as leukocytosis and fibrosis[9]. Therefore, MPN and particularly PMF patients are characterized by elevated levels of inflammatory cytokines, including interleukin (IL)−1ß, IL-6, tumor necrosis factor alpha (TNF-α), and interferon alpha (IFN-α)[10–16].

The current generation of JAK2 inhibitors is designed to compete for the ATP-binding site in the tyrosine kinase domain of JAK2[17]. Ruxolitinib is a potent JAK2/JAK1 specific inhibitor approved by the food & drug administration (FDA) and European Medicines Agency (EMA) for the treatment of PMF. The primary clinical benefit of JAK2 inhibitors in PMF patients is a reduction in spleen size and alleviation of constitutional symptoms resulting in a significant improvement in the quality of life[18–20]. However, no significant improvement of bone marrow fibrosis could be demonstrated so far, and the effect on JAK2-V617F allelic burden is modest[21]. Additionally, the beneficial clinical effects could also be demonstrated in JAK2-V617F⁻ PMF patients[19]. Thus, it remains unclear whether the drug primarily blocks the proliferation of the malignant clone or exerts its effects via non-malignant hematopoietic and/ or bone marrow niche cells.

In the present study, by deploying ruxolitinib-resistance JAK2-V617F mouse models, we show that ruxolinitb primarily acts by inhibiting non-oncogenic JAK signaling, reducing inflammatory cytokines. These findings suggest that ruxolitinib's clinical benefit derives largely from inhibiting non-oncogenic signaling.

## Results

### Identification of two JAK2-V617F mutations that confer high resistance to ruxolitinib

Ruxolitinib-resistant JAK2-V617F point mutations have not been described in MPN patients so far. In order to generate a ruxolitinib-resistant JAK2 variant, we introduced a kinase domain gate keeper residue mutation of methionine 929 to isoleucine which has been shown as highly resistant towards imatinib in the case of BCR-ABL. However, JAK2-M929I did not show any resistance to ruxolitinib as reported previously[22–24]. We therefore deployed a drug screening strategy using N-ethyl-N-nitrosourea (ENU) pretreatment, which we and others have used successfully for the identification of TKI resistant mutants[25]. At 8 µM ruxolitinib, we identified ruxolitinib resistant clones of the murine pro-B cell line Ba/F3. Sequencing of the JAK2 kinase domain of the resistant clones revealed point mutations in leucine residues L902Q and L983F at high abundance (44% and 18%, respectively). Additionally, some of the resistant clones displayed compound exchanges L902Q + R938E, L902Q + R947Q, L902Q + E1028K and L983F + N959H at lower frequency (Supplementary Fig. 1)[26]. Treatment of murine Ba/F3 cells stably expressing JAK2-V617F or JAK2-V617F + L902Q with ruxolitinib showed STAT5 activation at up to 4 µM ruxolitinib for JAK2-V617F + L902Q, whereas complete inhibition of STAT5 phosphorylation was observed at 250 nM ruxolitinib for JAK2-V617F cells (Fig. 1A and Supplementary Fig. 2A–D). Furthermore, different concentrations of ruxolitinib treatment confirmed that JAK2-V617F + L902Q is highly resistant towards ruxolitinib with an IC50 of around 6 µM, compared to 250 nM for JAK2-V617F cells (Fig. 1B). We also characterized the L983F mutation and found it even more resistant to ruxolitinib, as indicated by an IC50 of around 44 µM (Fig. 1C). STAT5 activation was still detected at up to 40 µM ruxolitinib (Fig. 1D and Supplementary Fig. 3), indicating that despite ruxolitinib oncogenic JAK2-V617F + L938F downstream signaling occurred. In addition to STAT5, we also examined other signaling pathways activated by oncogenic JAK2, including STAT3, and ERK1/2. We observed that in both JAK2-V617F + L902Q and JAK2-V617F + L983F these proteins remained activated at up to 4 µM ruxolitinib (Supplementary Fig. 4A, B).

To test whether L902Q alters the intrinsic kinase activity of JAK2, we measured the proliferation of transduced Ba/F3 cells without IL-3. As previously reported[3,27], expression of JAK2-V617F confers growth factor independent proliferation to Ba/F3 cells, similar to JAK2-V617F + L902Q whereas JAK2 WT and JAK2-L902Q alone did not induce cytokine independent proliferation (Supplementary Fig. 5A). Correspondingly, we found that JAK2-V617F + L902Q induces STAT5

activation similar to JAK2-V617F, while JAK2-L902Q alone failed to activate STAT5 (Supplementary Fig. 5B–D). These results indicate that L902Q does not alter the intrinsic kinase activity of JAK2. In addition to Ba/F3 cells, we also transduced primary bone marrow cells with JAK2-V617F, JAK2-V617F + L902Q, JAK2-V617F + L983F, and empty vector (MiG) and measured colony forming ability in the absence of cytokines in methylcellulose medium. Similar to JAK2-V617F, both JAK2-V617F + L902Q and JAK2-V617F + L983F displayed similar numbers of BFU-E, CFU-E, and CFU-GM (Supplementary Fig. 6A–C). These findings suggest that the drug resistant variants did not alter the cellular outgrowth phenotype of primary bone marrow compared to JAK2-V617F alone.

To understand the functional significance of Leu residues at positions 902 and 983 in generating ruxolitinib resistance, we modeled their interaction using a bioinformatics tool (Fig. 1E). Ruxolitinib fits well into the ATP-binding pocket of the kinase domain of JAK2, with 91% of its solvent-accessible surface area buried in the complex. The pyrrolopyrimidine moiety and pyrazol ring present in ruxolitinib form interactions with several amino acid residues of the JAK2 ATP-binding pocket. The main interactions are the carbonyl group of E930 accepting a hydrogen bond from the pyrrol ring and the amide group of L932 in the hinge region forming a hydrogen bond with the pyrimidine ring. Ruxolitinib is bound by numerous hydrophobic interactions with residues in the binding pocket such as L855, V863, A880, V911, M929, L932, and L983. L902 does not interact directly with ruxolitinib, but is close to the binding pocket and substitution with glutamine significantly disrupts binding of the inhibitor. The L983F mutation disrupts important hydrophobic interactions (e.g., Ala880, Val911, Met929) with the pyrrolopyrimidine moiety and induces aromatic-aromatic interactions between the new phenyl ring and the pyrrol and pyrazol rings. Substitution of phenylalanine at this position results in the loss of two critical hydrogen bonds between Glu930 and Leu932, which are present in all other JAK2 variants, explaining the weak affinity of ruxolitinib towards L983F (Fig. 1E). Thus, the structural modeling data support our finding that JAK2-L902Q and JAK2-L983F are highly resistant to ruxolitinib.

### JAK2-V617F + L902Q induces an MPN-like disease similar to JAK2-V617F in a retroviral bone marrow transplantation mouse model

To investigate the impact of JAK2-L902Q on the disease phenotype in a murine BM transplantation model, we transduced JAK2-V617, JAK2-V7617F + L902Q, and empty vector (MiG) into BM cells derived from Balb/cAan WT mice (Janvier Labs, Le Genest-Saint-Isle, France) and injected them into lethally irradiated Balb/cAan WT recipient mice. Mice receiving BM cells ectopically expressing JAK2-V617F + L902Q develop an MPN with increased hematocrit (HCT), Hemoglobin (HGB), reticulocytes, white blood cell (WBC), and RBC in the peripheral blood, similar to mice receiving JAK2-V617F transduced BM (Fig. 2A-C, Supplementary Fig. 7A, B). In addition to elevated blood counts, both groups developed a profound splenomegaly with a 4-fold increase in median spleen weight compared to empty vector control mice (Fig. 2D). Organ analysis revealed that both JAK2-V617F and JAK2-V617F + L902Q transplanted mice showed an increase in CD11b⁺Gr-1⁺ cells in BM, PB and spleen (Fig. 2E, Supplementary Fig. 7C, D, E). Microscopically, spleens derived from JAK2-V617F and JAK2-V617F + L902Q mice showed a marked infiltration by hematopoietic cells with a left shifted granulopoiesis, erythropoiesis, and moderately increased megakaryopoiesis. JAK2-V617F and JAK2-V617F + L902Q transplanted mice showed similar grade II myelofibrosis 60 days after transplantation (Fig. 2F; Supplementary Fig. 8 for enlarged view). This indicates that the resistance mutation L902Q did not enhance the MPN phenotype per se.

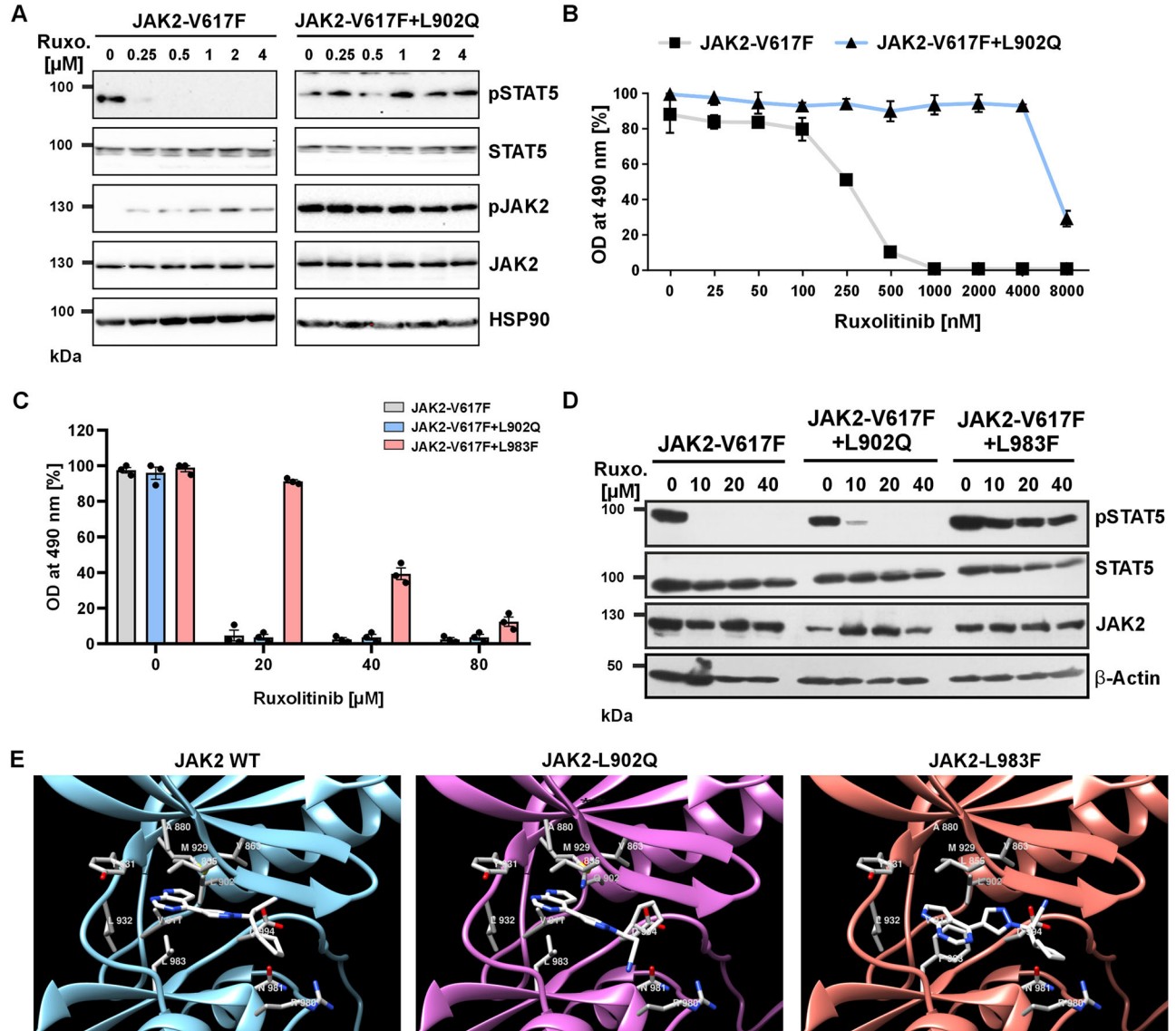

**Fig. 1 | A chemical mutagenesis screen identifies of ruxolitinib resistant JAK2 mutations L902Q and L983F. A** Immunoblot analysis of JAK2-V617F and JAK2V617F + L902Q expressing Ba/F3 cells in presence of indicated concentrations of ruxolitinib. The samples derive from the same experiment but different gels for pJAK2, total JAK2, HSP90, another for pSTAT5, total STAT5 were processed in parallel. Uncropped images are provided as a source data file. **B** MTT incorporation assay of JAK2-V617F and JAK2V617F + L902Q expressing Ba/F3 cells in presence of indicated ruxolitinib concentrations. Data represent mean ± SEM. Three independent experiments were performed. One representative experiment ($n = 3$) was shown. **C** MTT incorporation assay of JAK2-V617F, JAK2V617F + L902Q, and JAK2-V617F + L983F expressing Ba/F3 cells in presence of indicated ruxolitinib concentrations. Data represent mean ± SEM. Three independent experiments were performed. One representative ($n = 3$ technical replicates) experiment was shown. **D** Immunoblot analysis ($n = 2$) of JAK2-V617F, JAK2V617F + L902Q and JAK2-V617F + L983F expressing Ba/F3 cells in presence of indicated concentrations of

ruxolitinib. The samples derive from the same experiment but different gels for pSTAT5, total STAT5, another for JAK" and β-actin were processed in parallel. Uncropped images are provided as a Source Data file. **E** The binding of ruxolitinib to the JH1 domain of JAK2 kinase was modeled using the SwissDock tool to understand the structural consequences of mutations conferring resistance to ruxolitinib. Ruxolitinib fits well into the ATP-binding pocket of JAK2 WT. 91% of its solvent accessible surface area is buried in the complex. The drug is held by numerous hydrophobic interactions with residues Leu 855, Val 863, Ala 880, Val 911, Met 929, Leu 932 and Leu 983 that line the binding pocket. Leu 902 does not directly interact with ruxolitinib, however it is close to the binding pocket and its mutation to Gln with a polar side chain significantly disturbs the binding of the inhibitor. While it is still almost completely buried, the propanenitrile and cyclopentyl moieties essentially exchange their positions leading to unfavorable interactions between the Asp 994 side chain and the cyclopentyl ring.

## Ruxolitinib treatment decreases disease activity and inflammatory cytokines in both ruxolitinib sensitive and resistant JAK2-V617F mediated MPNs

Previous studies showed that ruxolitinib treatment in PMF patients and MPN mouse models achieves strongly decreased spleen sizes and reduction of inflammatory cytokines levels[18–20]. In order to study the underlying mechanism, we treated mice transplanted with either ruxolitinib sensitive or resistant JAK2-V617F mutant cells with 60 mg/kg

ruxolitinib twice daily over a period of 30 days. Ruxolitinib significantly reduced spleen weight and size in JAK2-V617F + L902Q mice, similar to JAK2-V617F mice (Fig. 3A–C). Consistent with these results, ruxolitinib treatment also led to a decrease in the total WBC counts in both drug sensitive and resistant mutant mice (Fig. 3D). Ruxolitinib treatment significantly reduced the number of myeloid CD11b+Gr-1+EGFP+ (oncogene transduced) but also CD11b+Gr-1+EGFP- (not oncogene transduced) cells in JAK2-V617F + L902Q PB and spleens (Fig. 3E, F,

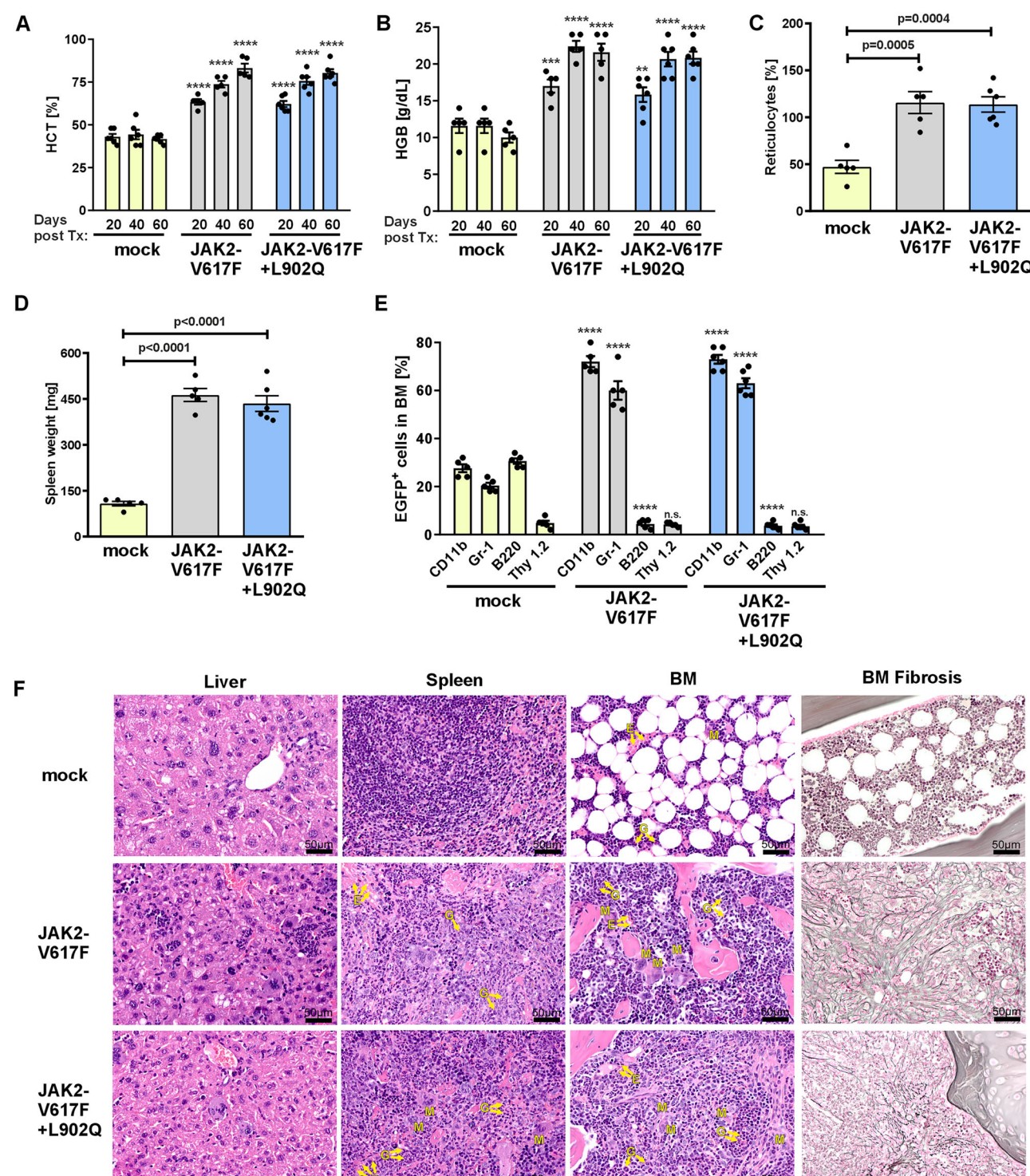

supplementary Fig. 9A, B). We also investigated whether ruxolitinib has an impact on myelofibrosis in JAK2-V617F mice. We detected no significant reduction of fibrosis in both ruxolitinib sensitive and resistant animals (Fig. 3G, H). These findings indicate that despite the highly resistant mutation that allow STAT5 signaling in cells treated with ruxolitinib, the in vivo treatment induces a reduction in disease activity similar to the non-mutated mouse model. This indicates that other factors than oncogenic JAK2 signaling contribute to the disease. Such factors could be pro-inflammatory cytokines.

PMF patients display elevated inflammatory cytokine levels[10]. Similarly, the JAK2-V617F mice show significantly increased inflammatory cytokines including tumor necrosis factor (TNF)-alpha, IL-6, and chemokine (C-C motif) ligand 2 (CCL2/MCP-1)[14]. Ruxolitinib treatment significantly reduced the levels of TNF-α, IL-6, CCL2/ MCP-1, interferon-beta (INF-ß), and IL-27 in peripheral blood sera of JAK2-V617F + L902Q mice, similar to JAK2-V617F mice (Fig. 4A–E). These cytokines are believed to play a major role in myelofibrosis. As we noticed significant down regulation of inflammatory cytokine levels in the serum of both sensitive and resistant models, we sorted EGFP+ cells from the bone marrow and spleen of JAK2-V617F and JAK2-V617F + L902Q mice and measured the inflammatory cytokine levels by microarray-based analysis. We did not observe any significant

**Fig. 2 | JAK2-V617F + L902Q mice display MPN phenotype and myelofibrosis similar to JAK2-V617F mice. A–D** JAK2-V617F + L902Q mice ($n = 6$) display increased **A** hematocrit value (HCT). Data represent mean ± SEM. *P* value was calculated using two-way ANOVA test. (****$p < 0.0001$ mock vs JAK2-V617F mice at day 20, 40, and 60). (****$p < 0.0001$ mock vs JAK2-V617F + L902Q mice at day 20, 40 and 60). **B** Hemoglobin levels (HGB), Data represent mean ± SEM. *P* value was calculated using two-way ANOVA test. (***$p = 0.0009$ mock vs JAK2-V617F mice at day 20), (****$p < 0.0001$ mock vs JAK2-V617F mice at day 40 and 60). (**$p = 0.0068$ mock vs JAK2-V617F + L902Q mice at day 20), (****$p < 0.0001$ mock vs JAK2-V617F + L902Q mice at day 40 and 60. **C** Reticulocyte percentage. Data represent mean ± SEM. *P* value was calculated using one-way ANOVA test. **D** Spleen weight compared to empty vector (mock) transplanted mice ($n = 6$), similar to JAK2-V617F mice ($n = 5$). Data represent mean ± SEM. *P* value was calculated using one-way ANOVA test. **E** Flow cytometric analysis of BM lineage composition at day 60 showing increased percentage of EGFP⁺ CD11b⁺ and Gr-1⁺ cells in JAK2-V617F ($n = 5$) and JAK2-V617F +

L902Q mice ($n = 6$) compared to mock mice ($n = 5$). Data represent mean ± SEM. *P* value was calculated using two-way ANOVA test. ****$p < 0.0001$ mock CD11b vs JAK2-V617F and ****$p < 0.0001$ mock CD11b vs JAK2-V617F + L902Q; ****$p < 0.0001$ mock Gr-1 vs JAK2-V617F and ****$p < 0.0001$ mock Gr-1 vs JAK2-V617F + L902Q; ****$p < 0.0001$ mock B220 vs JAK2-V617F and ****$p < 0.0001$ mock B220 vs JAK2-V617F + L902Q. n.s: non-significant $p = 0.9681$ in mock Thy1.2 versus JAK2-V617F and $p = 0.8474$ mock Thy1.2 vs JAK2-V617F + L902Q; **F** Histopathologic H&E stainings of liver, spleen and BM from JAK2-V617F and JAK2-V617F + L902Q mice reveal hyperplastic, left-shifted myelopoiesis granulopoiesis (G), erythropoiesis (E), and moderately increased megakaryopoiesis (M) in bone marrow. Gomori reticular fiber staining shows marked myelofibrosis. Note the extramedullary hematopoiesis in liver and spleen secondary to BM myelofibrosis. In contrast the mock animal shows normal liver, spleen and BM. (all figures 400x). Source data is provided in the Source Data file.

reduction of cytokines in malignant (EGFP⁺) cells in the resistant model by ruxolitinib and only a reduction of TNF-α, but not IL-6, MCP-1, INF-ß, or IL-27 in the sensitive animals (Fig. 4F–J and Supplementary Fig. 10A–E). Thus, cytokine production by malignant cells did not seem to contribute significantly to the observed effects of ruxolitinib in our MPN models. In an additional experiment, mice were transplanted with EGFP⁺ JAK2-V617F vs. JAK2-V617F + L902Q cells alongside EYFP⁺ empty vector cells. Consistent with the previous cytokine data, intracellular flow cytometry revealed a significant downregulation of STAT5 phosphorylation in CD11b⁺1⁺EYFP⁺ granulocytes, but not in EGFP⁺ granulocytes from JAK2-V617F and JAK2-V617F + L902Q mice (Supplementary Fig. 11A–C). This suggests that the clinical benefit of ruxolitinib may primarily result from the inhibition of cytokine production originating from non-malignant rather than malignant sources.

We also treated MiG-empty control animals with ruxolitinib and observed a significant reduction in spleen weight and length (Supplementary Fig. 12A, B). This data indicates some level of basic inflammation in these mice under steady-state conditions[28] which is reduced by ruxolitinib treatment. However, MiG-transplanted mice did not show an increase in WBC, HGB, or HCT in peripheral blood and ruxolitinib treatment did not result in any significant changes (Supplementary fig. 12C–G). In contrast to JAK2-V617F, the MiG control group showed elevated serum levels only for TNF-α and IL-27, while MCP-1, IL-6, INF-β, and IL-23 were low (Supplementary Fig. 12H). Ruxolitinib had no effect on none of these cytokines (Supplementary Fig. 12H).

### Ruxolitinib reduces disease activity independent of oncogenic signaling in a second highly ruxolitinib resistant JAK2-V617F mouse model

To validate our findings in a second model, we transplanted mice with JAK2-V617F + L983F, which is even more resistant to ruxolitinib (IC₅₀ around 44 μM, Fig. 1C). We observed that JAK2-V617F + L983F transplanted mice develop an MPN disease with increased HCT, HGB, myeloid cells, and WBC, similar to JAK2-V617F + L902Q and JAK2-V617F mice (Supplementary Fig. 13A–E). Ruxolitinib treatment of JAK2-V617F + L983F mice significantly reduced the spleen size and WBC counts (Fig. 5A, B) and the number of CD11b⁺Gr-1⁺ cells both in malignant and non-malignant populations (Fig. 5C, D). The fibrosis score both in sensitive and resistant models was not reduced by ruxolitinib (Fig. 5E, F).

Additionally, we measured the total number of peripheral blood B-cells following ruxolitinib treatment in both ruxolitinib resistant JAK2 models. We found that ruxolitinib treatment reduced the B-cell compartment of both EGFP⁺ and EGFP⁻ origin in JAK2-V617F + L902Q and JAK2-V617F + L983F similarly to JAK2-V617F (Supplementary Fig. 14A–D).

To characterize the pharmacological properties of ruxolitinib treatment in our model, we analyzed its serum abundance using mass spectrometry 3 h, 6 h, and 12 h after oral gavage. We detected a peak ruxolitinib serum concentration of 4.56 μM at the 3-h time point, whereas ruxolitinib was not detected 6 or 12 h after treatment (Fig. 5G, H). At these concentrations, JAK2-V617F is sufficiently blocked while the resistant mutants L902Q and L983F both retain activity.

### Ruxolitinib impairs inflammatory cytokine production by murine bone marrow stroma cells in vivo

Results thus far raised the question which cells are targeted by ruxolitinib since JAK2-V617F positive cells express a highly resistant kinase. Stromal cells have been shown to be among the main producers of inflammatory cytokines via JAK1/2 signaling[29,30]. To investigate the impact of stromal cells regarding reduced cytokine levels after ruxolitinib treatment, we analyzed global expression profiles of sorted mesenchymal stem cells (MSCs) and endothelial cells (ECs) from the BM of ruxolitinib vs. vehicle treated JAK2-V617F mice. We performed Metascape analysis[31] to acquire a comprehensive view of enriched biological pathways. Remarkably, all top eight pathways enriched in vehicle samples consisted of inflammation-related pathways, such as gene ontology (GO): inflammatory response or GO: cytokine production (Fig. 6A). We investigated these pathways in detail using gene set enrichment analysis (GSEA)[32,33] and found significant enrichment of these gene sets in both ECs and MSCs (Fig. 6B, C). We also detected significantly decreased expression of various inflammatory cytokines such as, TNF-α, IL-6, GM-CSF and leukemia inhibitory factor (LIF) (Fig. 6D). To validate the microarray data, we measured inflammatory cytokine levels in endothelial cells using intracellular FACS (Supplementary Fig. 15A, B). Consistent with the microarray findings, we observed significant reductions in TNF-alpha, and GM-CSF levels in endothelial cells from both JAK2-V617F and JAK2-V617F + L902Q mice (Fig. 6E, F).

These results show that ruxolitinib treatment significantly impairs cytokine production by bone marrow stroma cells in a murine MPN model, which are JAK2 WT.

### Ruxolotinib inhibits pro-inflammatory cytokine production from myeloid cells and T-cells of non-malignant origin

To further understand the mechanism of action of ruxolitinib on different hematopoietic compartments in MPN mouse models, we sorted hematopoietic cells of non-malignant origin from JAK2-V617F mice spleen and bone marrow. Analysis of cytokine expression levels suggested that inflammation-related pathways, such as inflammatory response or cytokine production were affected in non-malignant myeloid cells and T cells of spleen and bone marrow of JAK2-V617F mice. Inflammatory cytokines such as INF-γ, IL-10 and chemokines such as CCL-3, CCL-20, CXCL-3 in myeloid cells of non-malignant

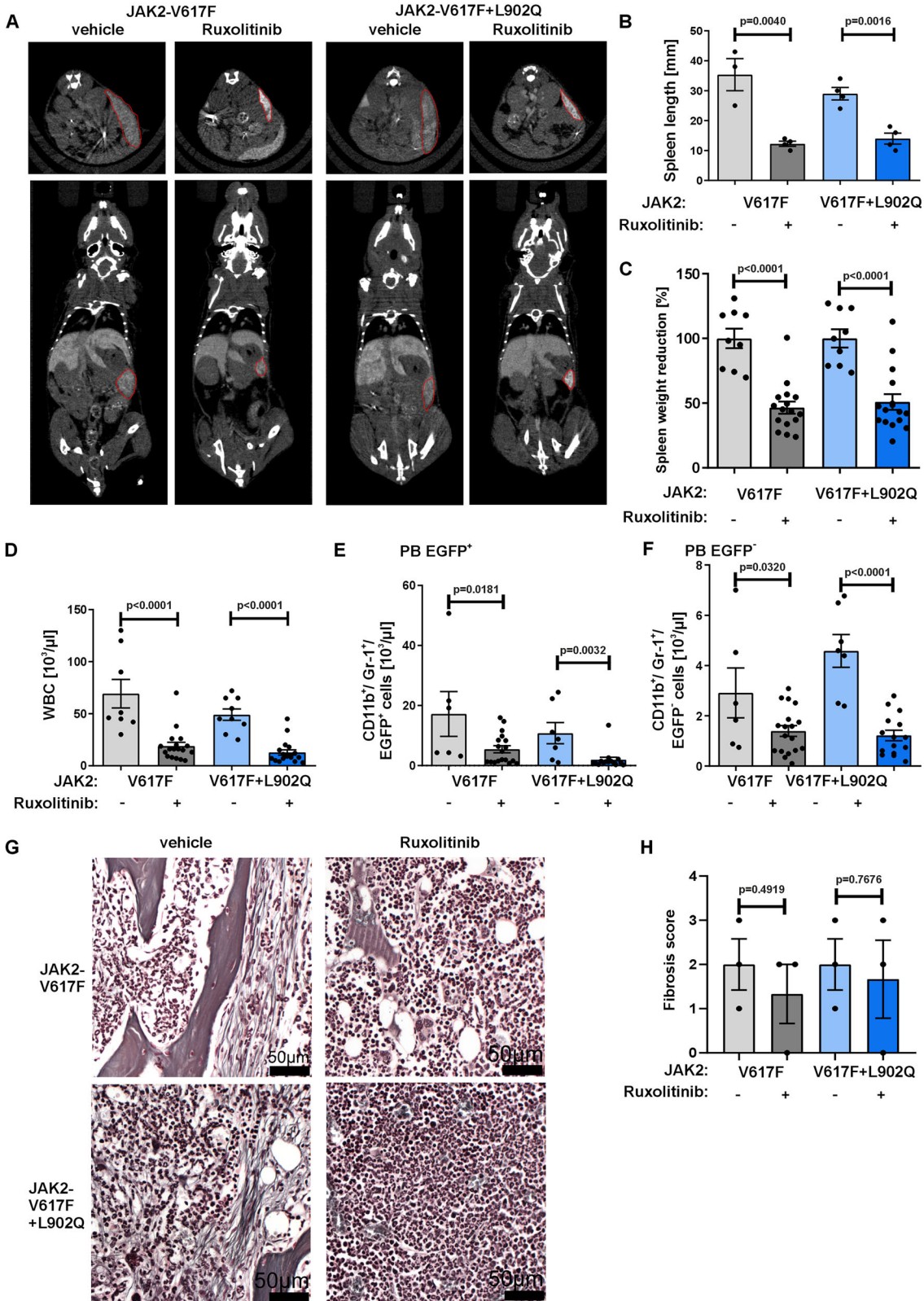

origin were significantly decreased after ruxolitinib treatment (Supplementary Fig. 16A). Likewise, analysis of T cells from splenocytes of non-malignant origin showed downregulation of chemokines such as CCL-20, CXCL-3, CXCL-9, and CXCL-10 (Supplementary Fig. 16B). These results indicate that ruxolitinib acts on the production of inflammatory cytokines and chemokines in non-malignant myeloid- and T-cells by inhibiting JAK-family kinases mediated signaling.

## JAK1 selective inhibition by itacitinib does not decrease spleen size or WBC in neither the ruxolitinib sensitive nor resistant mouse model

To test the contribution of JAK1 and JAK2 inhibition, we treated JAK2-V617F and JAK2-V617F + L902Q mice with either ruxolitinib (JAK1/JAK2 inhibitor) or the selective JAK1-inhibitor itacitinib[34]. Again, mice with either mutation developed an MPN like disease with increased RBC,

**Fig. 3 | Ruxolitinib treatment decreases spleen size and WBC in ruxolitinib sensitive and resistant JAK2-V617F transplanted mice. A** Representative CT scan images from JAK2-V617F and JAK2-V617F-L902Q mice treated with vehicle or ruxolitinib. Red borders indicate spleen size. **B** Statistical analysis of spleen length determined by CT scan imaging from JAK2-V617F and JAK2-V617F + L902Q mice treated with vehicle ($n = 3/4$, respectively) or ruxolitinib ($n = 4$). Data represent mean ± SEM. $P$ value was calculated using two-tailed Student's $t$ test. **C** Ruxolitinib treatment ($n = 16$) reduces spleen weight of both JAK2-V617F and JAK2-V617F + L902Q mice at day 60, compared to vehicle treated mice ($n = 9$). Data represent mean ± SEM. $P$ value was calculated using two-tailed Student's $t$ test. **D** Statistical analysis of WBC at day 60 from JAK2-V617F and JAK2-V617F + L902Q mice treated with vehicle ($n = 8/9$, respectively) or ruxolitinib ($n = 18$). Data represent mean ± SEM. $P$ value was calculated using two-tailed Student's $t$ test. (**E** + **F**) Flow cytometric analysis of **E** EGFP+ and **F** EGFP- CD11b+/Gr-1+ cells in peripheral blood (PB) of JAK2-V617F and JAK2-V617F + L902Q mice treated with vehicle ($n = 6/7$ respectively) or ruxolitinib ($n = 18/15$, respectively). Data represent mean ± SEM. $P$ value was calculated using two-tailed Student's $t$ test. **G** Representative Gomori staining images showing reduced myelofibrosis in ruxolitinib treated animals. **H** Histopathologic scoring of fibrosis in JAK2-V617F ($n = 3$) and JAK2-V617F + L902Q mice ($n = 3$) shows slight reduction of fibrosis in ruxolitinib vs. vehicle treated animals. Data represent mean ± SEM. $P$ value was calculated using two-tailed Student's $t$ test. Source data is provided in the Source Data file.

HCT, HGB, WBC and myeloid cells (Fig. 7). While ruxolitinib in both JAK2 mutations confirmed the results described before, itacitinib failed to show any effects on WBCs, myeloid cells, spleen weight or length (Fig. 7A–D). Additionally, the reduction in the number of malignant and non-malignant CD11b+Gr-1+ cells elicited by ruxolitinib was not observed for itacitinib (Fig. 7E, F) suggesting that JAK1 inhibition alone does not achieve a relevant therapeutic response in MPN mice. This data is in line with a clinical trial investigating itacitinib in myelofibrosis patients, where a spleen size reduction of only 14% after 12 weeks treatment was observed as compared to 50% by ruxolitinib[35]. Taken together our results indicate that the mechanism of action for ruxolitinib in MPN is based on the inhibition of cytokines in non-hematopoietic cells such as stromal cells as well as non-malignant hematopoietic cells.

### Susceptibility towards JAK inhibition in JAK2-V617F mice depends on the nonmalignant hematopoietic cells

In our in vivo models, we found that ruxolitinib was preferentially effective in suppressing the activity of the non-malignant cell population. To corroborate this concept, we cloned the JAK2-L902Q variant without the oncogenic mutation. We observed that the JAK2-L902Q variant had an IC$_{50}$ of >4 µM to block proliferation, compared to 1.5 µM for JAK2-wild type cells (Supplementary Fig. 17A–F). Next, we transplanted EGFP+-sorted oncogenic JAK2-V617F together with EYFP+-sorted ruxolitinib resistant JAK2-L902Q bone marrow cells vs. EYFP+-sorted empty vector bone marrow. Using this strategy, we induced an MPN-like disease in the mice, as evidenced by an increase in HGB and HCT (Supplementary Fig. 18A, B). However, ruxolitinib treatment did not alter HGB, or HCT in JAK2-V617F/JAK2-L902Q mice compared to JAK2-V617F/empty mice (Supplementary Fig. 18C, D). Intracellular FACS staining for STAT5 phosphorylation showed a significant reduction in CD11b+Gr-1+ EYFP+ granulocytes of JAK2-V617F/empty mice (Supplementary Fig. 18E). However, no significant reduction was observed in the JAK2-V617F/JAK2-L902Q mice (Supplementary Fig. 18E). These mice also did not show any significant reduction in spleen size following ruxolitinib treatment (Supplementary Fig. 18F). This data suggests that the therapeutic effects of ruxolitinib are primarily driven by inhibition of JAK2 signaling in nonmalignant hematopoietic cells.

Taken together, using the bone marrow transplantation method, we demonstrate that the therapeutic benefits of ruxolitinib in MPN mice are mainly mediated by the inhibition of cytokine-mediated signaling in non-malignant hematopoietic cells and bone marrow stromal cells rather than in malignant cells.

### Discussion

The JAK2-V617F mutation is a major driver in MPNs comprising PV, ET and PMF[3,5–7]. JAK2 inhibitors, such as ruxolitinib show remarkable clinical activity in myelofibrosis patients, irrespective of JAK2 mutational status[18–20]. PMF patients who undergone ruxolitinib therapy show reduction of spleen size as well as inflammatory cytokine levels and significant improved quality of life[19]. However, ruxolitinib does not significantly improve BM fibrosis and the decrease of mutant allelic burden is modest[19,36].

In the COMFORT-II trial about 30% of the patients had a reduction in the JAK2V617F allelic burden of more than 20%[37]. For PV, in the RESPONSE trial, even a more pronounced reduction in allelic burden could be demonstrated[38,39]. However, there was no correlation between the reduction in allelic burden and clinical benefit. Therefore, most practitioners do not consider allelic burden to be a meaningful biomarker in MPN treated with ruxolitinib. If allelic burden does not reflect the clinical benefit of ruxolitinib therapy, it is tempting to speculate that the clinical benefits of ruxolitinib may not be achieved by directly suppressing the JAK2-V617F positive malignant clone, but rather by its inhibitory effect on cytokine signaling in non-malignant cells. Other observations support this hypothesis: (a) ruxolitinib has clinical activity in PMF patients independent of the presence of JAK2-V617F, and (b) ruxolitinib resistance is not associated with the selection of JAK2-V617F secondary resistance mutations. The latter is a strong argument that the kinase inhibitor does not put a strong selection pressure on JAK2-V617F positive clones, since secondary resistance mutations are a commonly observed resistance mechanism in malignancies where a kinase inhibitor clearly blocks oncogene-transformed cells. Therefore, it was speculated that ruxolitinib may act more like an anti-inflammatory drug such as cortisone in PMF and not like a specific oncogene kinase inhibitor such as imatinib or osimertinib. In fact, ruxolitinib recently also achieved approval for graft-versus-host-disease after allogenic transplantation due to its anti-inflammatory activity[40].

As it is not possible to clearly delineate the mechanisms of action of ruxolitinib in a clinical setting, we generated a JAK2-V617F MPN mouse model, expressing a JAK2-V617F variant, which is not blocked by ruxolitinib. This would allow to distinguish between the inhibitory effects of ruxolitinib on JAK2-V617F oncogenic signaling and normal JAK1/JAK2 signaling. Previously, we and others have established JAK2-V617F driven MPN mouse models by using retrovirus, transgenic mice, and patient derived xenografts (PDX)[7,14,27,41–43]. Here, we used a retroviral syngeneic mouse model. Since no JAK2-V617F resistance mutations in patients have been described so far, the first step was to identify highly ruxolitinib resistant JAK2V617F mutations. In an in vitro resistant screen, we identified two highly ruxolitinib resistant JAK2 mutations, JAK2-L902Q (IC$_{50}$: 6 µM) and JAK2-L983F (IC$_{50}$: 44 µM). Our findings are in line with three studies independently reporting JAK2 point mutations conferring resistance towards JAK2 inhibitors in vitro[22,24,44]. However, alteration of gate keeper residue (M929I) in full length JAK2 did not yield ruxolitinib resistance in contrast to another study using TEL-JAK2 where the authors suggested that the gate keeper residue might be involved in resistance[23]. This discrepancy might be based on the different modes of activation for full length JAK2 and TEL-JAK2.

Using the highly ruxolitinib resistant JAK2-V617F + L902Q and JAK2-V617F + L983F mutations we could establish an MPN disease in mice which was similar in onset and phenotype to the original

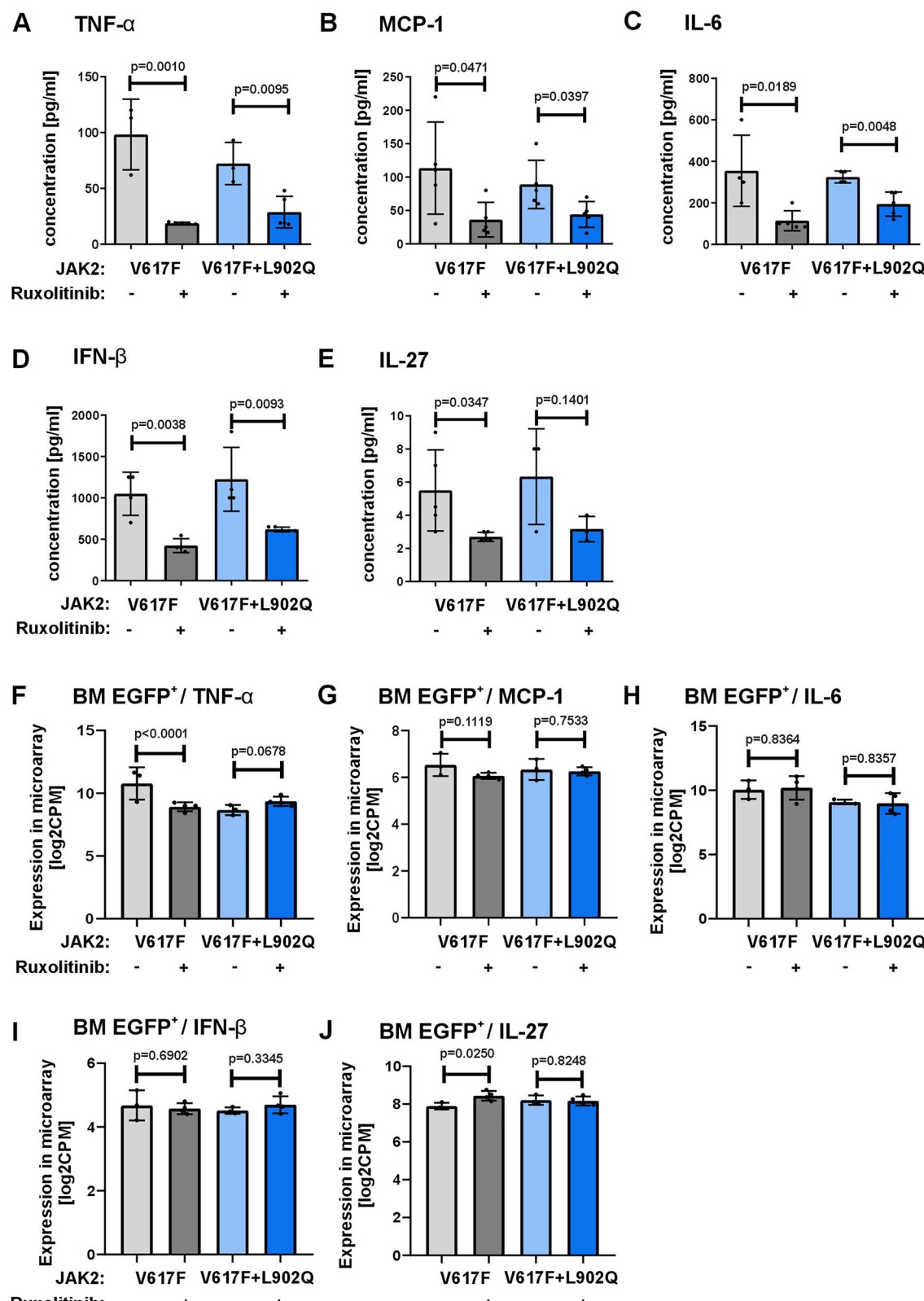

JAK2V617F model. We performed ruxolitinib treatment of twice-daily 60 mg/kg gavage, as used in several other studies[21,45]. However, the ruxolitinib serum kinetics after treatment have not been studied in mice before. Since serum kinetics of ruxolitinib in mice are crucial in our experimental setting, we determined the half-life of ruxolitinib in mice after twice daily treatment via gavage. We could demonstrate that ruxolitinib is short-lived in mouse

serum with a peak concentration of around 4 μM 3 h after gavage, which is well below the $IC_{50}$ of both JAK2-V617F + L902Q and JAK2-V617F + L983F. We therefore are confident that the ruxolitinib mediated effects in the resistant mouse models are not due to excess drug concentrations.

In the present study, we demonstrate that mice harboring highly ruxolitinib resistant JAK2-V617F mutants respond to ruxolitinib

**Fig. 4 | Ruxolitinib treatment decreases inflammatory cytokine levels in sera of ruxolitinib sensitive and resistant JAK2-V617F transplanted mice.** Serum levels of tumor necrosis factor alpha (TNF-α) (**A**), interleukin-6 (IL-6) (**B**), monocyte chemoattractant protein 1 (MCP-1)/chemokine (C-C motif) ligand 2 (CCL2) (**C**), interferon-beta (IFN-β) (**D**), and interleukin-27 (**E**) determined in ruxolitinib vs. vehicle treated JAK2-V617F and JAK2-V617F + L902Q mice ($n = 5$). Data represent mean ± SEM. *P* value was calculated using two-tailed Student's *t* test. **F**–**J** CD11b[+] granulocytes of EGFP[+] cells from bone marrow of JAK2-V617F and JAK2-V617F +

L902Q mice were isolated from vehicle and ruxolitinib treated groups. Microarray analysis was performed on these cells for cytokine landscape. Inflammatory cytokines such as of tumor necrosis factor alpha (TNF-α) (**F**), IL-6 (**G**), MCP-1/CCL2 (**H**), IFN-β (**I**), and IL-27 (**J**) were depicted from bone marrow (BM) compartment of the ruxolitinib ($n = 4$) vs. vehicle ($n = 3$) treated JAK2-V617F and JAK2-V617F + L902Q mice. Data represent mean ± SEM. Adjusted *p* value based on Benjamini-Hochberg Step-Up FDR-controlling Procedure. Source data is provided in the Source Data file.

treatment with a reduction in spleen size and WBC similar to mice harboring ruxolitinib sensitive JAK2-V617F, suggesting that ruxolitinib acts mainly on non-oncogenic JAK1/JAK2 signaling. Furthermore, we also detected a strong reduction of inflammatory cytokines such as TNF-α, IL-6, MCP-1, INF-β, and IL-27 in the serum of both ruxolitinib sensitive and ruxolitinib resistant mice. Additionally, we performed expression and flow cytometric analysis of inflammatory cytokines in different stromal cells and malignant and nonmalignant hematopoietic cells and found that ruxolitinib treatment reduces levels of cytokines in MSCs, ECs, and also in nonmalignant myeloid and T cells. These results suggest that the source of inflammatory cytokines in MPN originates from various cells including stroma cells and nonmalignant hematopoietic cells. These results are in line with previous findings that non-malignant cells are involved in the production of inflammatory cytokine levels in JAK2-V617F mediated MPNs[46]. Several reports within the last years have investigated the critical (and causative) role of inflammation, in particular cytokines, in MPN[47–49]. The contribution of cytokines produced by stromal cells to MPN disease induction has been the subject of numerous recent studies: IL-6 has been shown as a potential stimulator of angiogenesis in the tumor microenvironment and has been demonstrated to enhance proliferation and migration of endothelial cells[50–52]. TNF-alpha levels are significantly elevated in BCR-ABL positive CML[53,54] and other MPNs including PV, ET, and PMF[10,11,14]. Fleischman et al. demonstrated that TNF-alpha levels are elevated in JAK2-V617F positive cells and found that TNF-alpha reduces colony formation in normal hematopoietic cells, whereas JAK2-V617F positive progenitor cells were resistant to TNF-alpha[55]. Additionally, it was shown that JAK2-V617F reduces nestin[+] MSCs in the BM via IL-1ß production[42]. In fact, ruxolitinib was shown to reduce many but not all cytokines to normal levels[19,21,56]. This can be explained by the fact—as shown in the present study—that cytokines are a product from mutant hematopoietic cells and non-mutant hematopoietic cells and stromal cells.

Together, our results suggest that the clinical benefits seen in ruxolitinib treated MPN mice is due to the inhibition of non-oncogenic JAK signaling leading to the downregulation of inflammatory cytokines in various cells. Experiments with selective JAK1 or JAK1/JAK2 inhibitors indicate that the majority of the therapeutic effects is mediated by non-oncogenic JAK2 inhibition.

Our results suggested that non-oncogenic JAK2 inhibition in the malignant clone did not contribute significantly to the reduction of inflammatory cytokines in our MPN model. It is important to consider that by using a retroviral overexpression model, endogenous JAK2 is still expressed in the malignant cells. In MPN patients, JAK2-V617F can be either heterozygous (80%) or homozygous (20%)[3–5]. Our mouse model recapitulates the heterozygous situation by retaining the endogenous JAK2. To address the homozygous situation in a mouse model would require a homozygous knock-in into the endogenous JAK2. However, we would not expect different results in such a rather laborious approach.

Fibrosis is considered a disease hallmark of PMF and is a complex reactive process occurring in several organs, such as spleen and BM. Various pro-angiogenic cytokines including FGF-b, IL-8, VEGF, HGF, PDGFR, TGF-beta, TNF-alpha, and OSM have been implicated in BM microenvironment alterations in MPN patients[57–64]. Despite the

reduction of inflammatory cytokine production in the serum and stromal cells of ruxolitinib treated animals, we could not detect significant improvement in BM fibrosis. This finding is in line with PMF patients undergoing ruxolitinib therapy, which also do not show significant reduction of BM fibrosis[19,36].

## Methods

### Ethics statement

All research was conducted in accordance with relevant ethical guidelines. All animal procedures were performed in accordance with institutional animal care guidelines. The procedures were reviewed and approved by the Animal Welfare Officers of the University Hospital Freiburg and the Animal Ethics Committee of the local government (Veterinärwesen, Gesundheitlicher Verbraucherschutz und Lebensmittelüberwachung, Regierungspräsidium Freiburg, Freiburg Germany) under the license number (Az: G-13/05, G-22/093, and G-24/003).

Mice were housed in a specialized caging system with autoclaved food and acidified water, at a 12-h light/12-h dark cycle at 20–24 °C and 45–65% humidity, at the University Hospital Freiburg, adhering to national and institutional guidelines for animal care. Animals were frequently checked for signs of disease such as hyperemia and blood values were frequently measured to assess disease progression. Animals were humanely euthanized when immobile or paralysis, showing signs of distress or relevant body weight loss (>20%). Mice were used between 8 and 10 weeks of age.

### Cell culture

Ba/F3 cells and NIH/3T3 cells were obtained from the German Resource Centre for Biological Material. They were maintained in RPMI 1640 medium (Thermo Fisher, Waltham, MA, USA) supplemented with 10% fetal calf serum (FCS) and 2 ng/ml murine interleukin-3 (IL-3; Peprotech, Hamburg, Germany). Phoenix E retroviral ecotropic packaging cells were provided by G. Nolan (Stanford University, USA). These cells and NIH/3T3 cells were cultured in DMEM (Thermo Fisher) with 10% FCS and 1% penicillin-streptomycin.

### DNA constructs

The MSCV-IRES-JAK2-V617F constructs have been described previously[27]. JAK2 mutations were introduced in the MSCV-EYFP-JAK2-V617F vector using the QuickChange mutagenesis kit (Stratagene, San Diego, CA, USA).

### Proliferation assay

Ba/F3 cell proliferation in the presence of ruxolitinib at the indicated times was measured by incubation in an MTS (3-(4,5-dimethylthiazol-2-yl)-5-(3-carboxymethoxyphenyl-2-(4-sulfophenyl)-2H-tetrazolium) substrate for 2 h. After 2 h, formazan absorbance was measured at 490 nm (CellTiter 96; Promega, Madison, WI) using a cell plate reader. Measurements were conducted in triplicates after 48 and 76 h of culture without cytokines, as previously described[27].

### Western blot

Ba/F3 cells were treated with the indicated concentration of ruxolitinib for a period of 2.5 h. Cells were lysed using cell lysis buffer,

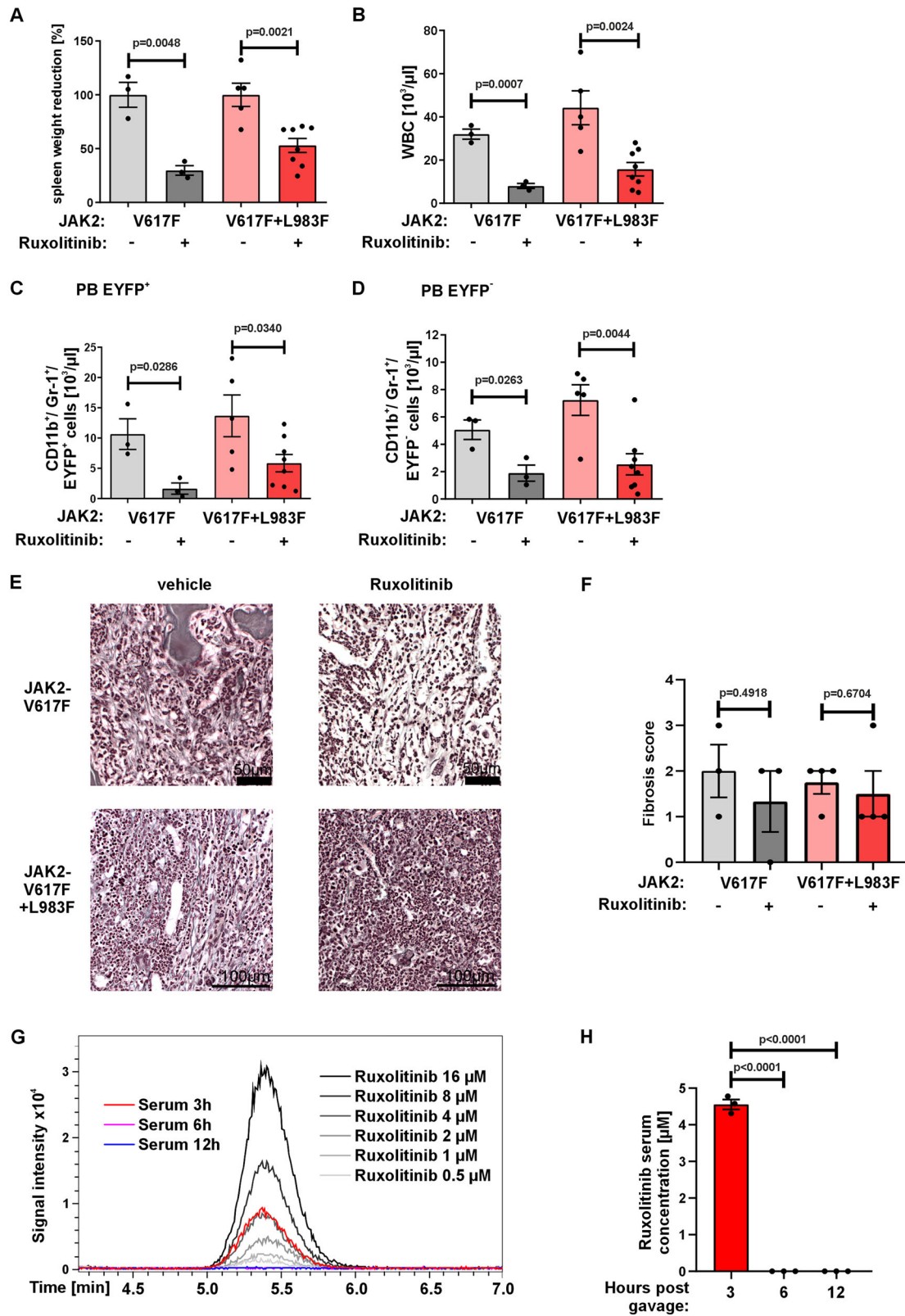

lysates were subjected to sodium dodecyl sulfate–polyacrylamide gel electrophoresis (SDS-PAGE), and immunoblotting were performed as described previously[65]. Bands were visualized using the enhanced chemoluminescence system (Amersham, Braunschweig, Germany). STAT5 (G-2) and pJAK2 (21870-R) antibodies were purchased from Santa Cruz Biotechnology (Santa Cruz, CA, USA). JAK2 antibody (D2E12 XP[R]) and pSTAT5, pSTAT3, STAT3, pAkt, Akt, pERK, ERK, HSP90, and GAPDH antibodies were purchased from Cell Signaling Technology (Danvers, MA, USA). All the antibodies were prepared in 1:1000 dilution in BSA or milk. Phospho-antibodies were prepared in BSA, total protein antibodies were prepared in milk.

Uncropped scans of the blots can be found at the online Source Data.

**Fig. 5 | Ruxolitinib treatment decreases spleen size and WBC in a highly rux-olitinib resistant JAK2-V617F + L983F mouse model.** Decreased **A** spleen weight and **B** white blood cells (WBC) in ruxolitinib vs. vehicle treated JAK2-V617F (*n* = 3) and JAK2-V617F + L983F mice (*n* = 5/8, respectively). Data represent mean ± SEM. *P* value was calculated using two-tailed Student's *t* test. Decreased EYFP⁺ (**C**) and EYFP⁻ (**D**) CD11b⁺/ Gr-1⁺ cells in ruxolitinib vs. vehicle treated JAK2-V617F (*n* = 3) and JAK2-V617F + L983F mice (*n* = 5/8, respectively). Data represent mean ± SEM. *P* value was calculated using two-tailed Student's *t* test. **E** Representative Gomori staining images showing reduced myelofibrosis in ruxolitinib treated JAK2-V617F and JAK2-V617F + L983F animals compared to vehicle. **F** Histopathologic scoring of fibrosis in JAK2-V617F (*n* = 3) and JAK2-V617F + L983F (*n* = 4 for vehicle, *n* = 5 for ruxolitinib treatment) mice shows slight reduction of fibrosis in ruxolitinib vs. vehicle treated animals. Data represent mean ± SEM. *P* value was calculated using two-tailed Student's *t* test. **G** Representative mass spectrometry analysis of three independent mouse serum samples (*n* = 3) at the indicated time points after oral gavage of 60 mg/kg ruxolitinib. Ruxolitinib standards were included at concentrations from 16 μM to 0.5 μM. **H** Statistical analysis of mass spectrometric measurements of three independent mice (*n* = 3) of ruxolitinib serum concentration at the indicated time points after oral gavage. At the 3-h time point, a maximum concentration of 4.56 μM was detected. Data represent mean ± SEM. *P* value was calculated using one-way ANOVA test. Source data is provided in the Source Data file.

### Generation of drug-resistant variants
The generation of drug resistant clones was described previously[25]. Briefly, Ba/F3 MSCV-EYFP-JAK2-V617F cells were pretreated with N-ethyl-N-nitrosourea (ENU) and cultured in 96 well plates at a density of $4 \times 10^5$ cells/well at 8 μM ruxolitinib. Visible colonies were picked, expanded, and analyzed. Inhibitor-resistant sublines were cultured at 8 μM ruxolitinib.

### RNA isolation and cDNA synthesis
Total RNA was extracted using TRIzol reagent (Invitrogen, Carlsbad, CA, USA) or RNeasy Plus Micro Kit (QIAGEN, Hilden, Germany) and transcribed into cDNA using the RevertAid First Strand cDNA Synthesis Kit (Thermo Scientific).

### Primers
JAK2 kinase domain RT-PCR primers: JAK2 RT–KD for gaaaatgacatgttaccaaatatg and JAK2 RT-KD rev ggagtaaacaaactgttaaag. Kinase domain sequencing primers: ctagggtttttctggtgcctttgaag and gggcgttg atttacattattgttcc. The primers used for site directed mutagenesis: JAK2-L902Q for: gggaaattgaaatccagaaatccctacagcatg, JAK2-L902Q rev: catgctgtagggatttctggatttcaatttccc, JAK2-L983F for: gcaacgagaaatatatt gtggagaacgag, JAK2-L983F rev: ctcgttctccacaaatatatttctcgttgc.

### Bone marrow transduction and transplantation
Bone marrow transduction and transplantation was performed as previously described[66]. Briefly, $2.5 \times 10^6$ Phoenix E cells were transiently transfected with Lipofectamine 2000 (Invitrogen, Karlsruhe, Germany) and after 24 h, medium was replaced with DMEM containing penicillin and streptomycin. Retroviral stocks were then harvested at 36 and 48 h after transfection. Viral titers were measured by transducing $5 \times 10^4$ NIH/3T3 cells with 1:20, 1:200, or 1:500 dilutions of retrovirus in the presence of 4 μg/mL polybrene (Sigma) by calculating the colony forming units per ml [CFU/mL]) of the retroviral supernatant. Murine bone marrow was harvested from 6–8-week-old male Balb/cAan donor mice 4 days after injection of 150 mg/kg 5-fluorouracil (Medac, Wedel, Germany) and cultured overnight in DMEM (Gibco) with 20% ES cell FCS (Stem Cell Technologies (Vancouver, Canada) supplemented with 10 ng/mL mIL-3, 10 ng/mL mIL-6, 50 ng/mL mSCF (Peprotech). Bone marrow cells were infected using by spin-infection (1200 g, 32 °C, 90 min), i.e., by incubating with retroviral supernatant supplemented with growth factors and 4 μg/mL polybrene (Sigma). After four rounds of transduction, the bone marrow cells were washed with PBS then suspended in Hanks balanced salt solution (Sigma) and injected into the tail vein of lethally irradiated (800 rad) female Balb/cAan recipient mice. Transplanted animals were monitored for signs of disease by serial measurement of peripheral blood (PB) counts. All procedures were reviewed and approved by the Animal Care Committee of the University and the local authorities in Freiburg.

### Analysis of transplanted mice
HGB, HCT, platelets, and WBC counts were determined using an automated counter (SCIL vet abc, SCIL, Altorf, France). Reticulocytes were stained with brilliant cresyl blue solution (1%) and quantified by light microscopy (per 1000 erythrocytes). Numbers of transduced enhanced green fluorescent protein (EGFP) or enhanced yellow fluorescent protein (EYFP) positive cells in the peripheral blood of transplanted mice were determined by flow cytometry.

### Histopathology
Spleen and bone marrow specimens were fixed in buffered formalin (4% pH 7.4), decalcified in EDTA and stained with Hematoxylin and Eosin (H&E) and gomori's reticulin staining. Slides were viewed with a Zeiss Axioplan 2 microscope (40x/0.75NA Plan-Neofluar air objective). Images were acquired using a Zeiss Axiocam MRc 5 camera and were processed with Axiovision Rel 4.6 scanning software.

### Flow cytometric analysis and sorting
Flow cytometry was performed on BD LSRFortessa™ or Beckman CyAn™ ADP analyzers. Sorting of murine stroma cell populations or spleen and bone marrow cells was performed on a BD FACSAria™ III[66,67]. Antibodies used for lineage staining: CD45, CD11b, Gr-1, CD 90.2 and B220 (all eBioscience).

### Detection of ruxolitinib in serum by mass spectometry
Mouse serum ruxolitinib concentrations were measured using a validated liquid chromatography method (Column: Reprosil Pur Basic C18, 3 μ, 100 × 2 mm, gradient with formic acid/0.1 %/methanol) with mass spectroscopic detection (ruxolitinib cation m/z = 307.2, Bruker qTOF). Briefly, mice were treated twice daily with ruxolitinib at a concentration of 60 mg/kg and serum was isolated at 1 h, 3 h, and 6 h after ruxolitinib treatment. After deproteinisation with methanol, serum ruxolitinib concentrations were determined by mass spectroscopic detection using the standard ruxolitinib concentrations (0.5 μM, 1 μM, 2 μM, 4 μM, 8 μM, and 16 μM) as a reference.

### Isolation of mesenchymal stromal cells (MSCs) and endothelial cells
Mesenchymal stromal cells (MSC) and endothelial cells were isolated as mentioned in previous publications[67,68]. In brief, for stromal niche cell isolation, flushed femurs and tibias were crushed with mortar and pestle. Bone chips were washed several times in PBS until the chips were white. Endosteal stromal cells were released from the hematopoietic-depleted bone chips by digestion with 3 mg/mL collagenase type I, 0.5 mg/ml collagenase Type II, and 15 μg/mL DNAse dissolved in PBS for 1 h at 37 °C at 110 rpm. The stromal and the BM fraction were used in all subsequent analyses. For bone chips cells, we used CD45/TER-119/ CD3/ Gr-1/(lineage)−PE-Cy7 (1:1,000 each), CD31-Pacific Blue (PECAM-1) (1:100), Sca-1-APC-Cy7 (1:100), CD166-PE (ALCAM) (1:50). For the BM fraction we used CD45/TER-119/ CD3/Gr-1/(lineage markers)−PE-Cy7 (1:1,000 each) CD31-Pacific Blue (1:100), Sca-1-APC-Cy7 (1:100), CD140-PE (1:50). The stained cells were analyzed by FACS Vantage (BD Biosciences). Sorting of MSC and endothelial were performed on BD FACSAria™ III.

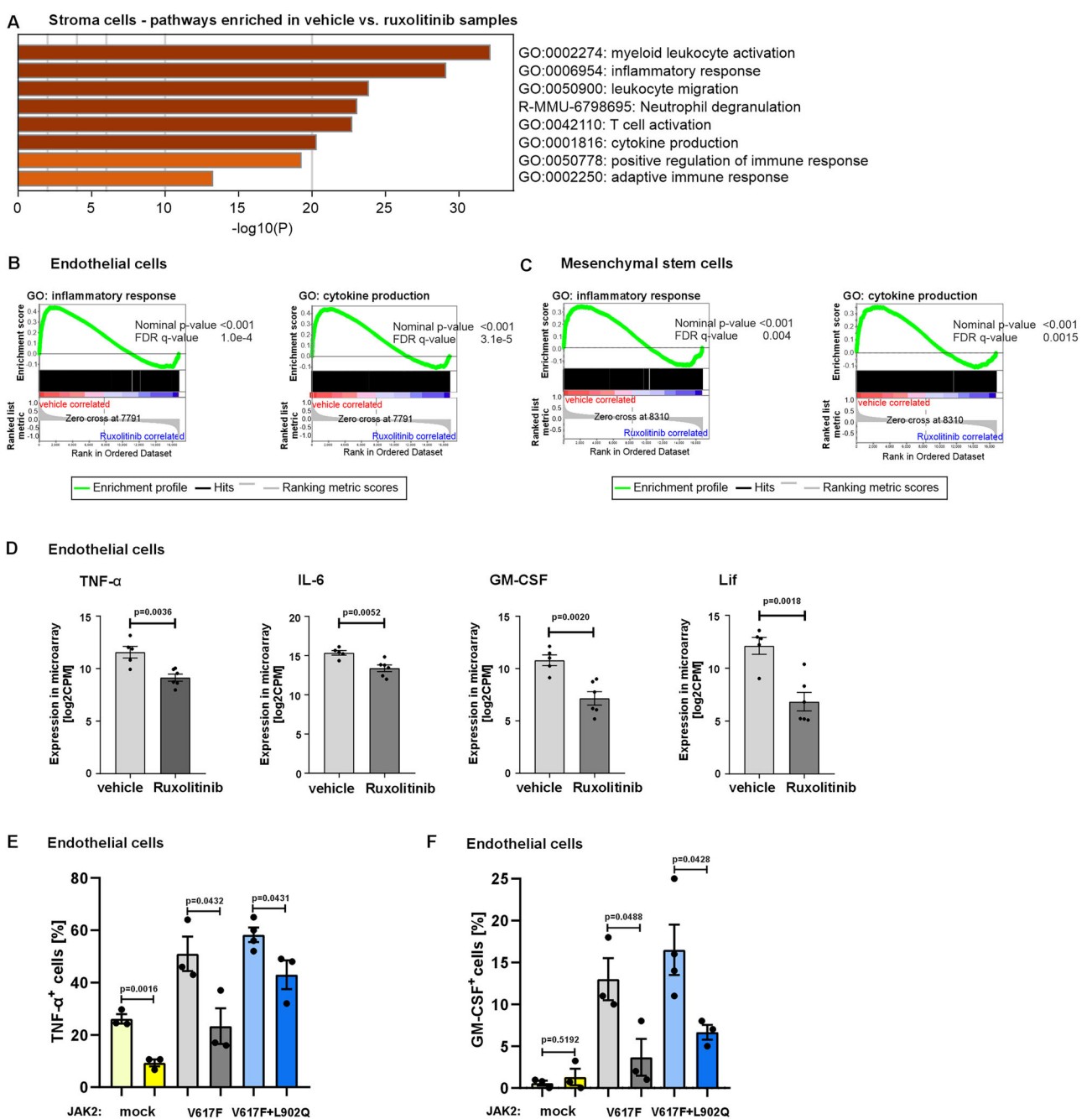

**Fig. 6 | Ruxolitinib treatment impairs inflammatory cytokine production in bone marrow stroma cells from JAK2-V617F mice. A** Metascape analysis of gene sets significantly enriched in stroma samples from vehicle ($n = 10$) vs. ruxolitinib ($n = 10$) treated JAK2-V617F mice. The top 8 gene sets are depicted. **B** Gene set enrichment analysis (GSEA) of endothelial cells from vehicle ($n = 5$) vs. ruxolitinib ($n = 6$) treated JAK2-V617F mice for the gene sets gene ontology (GO): inflammatory response and GO: cytokine production. **C** Gene set enrichment analysis (GSEA) of mesenchymal stem cells from vehicle ($n = 5$) vs. ruxolitinib ($n = 4$) treated JAK2-V617F mice for the gene sets GO: inflammatory response and GO: cytokine production. **B**, **C** False detection rate (FDR), $q$ value and nominal $p$ value were calculated using the Broad institute GSEA tool[33]. Statistical analyses were performed by

one-sided, Kolmogorov-Smirnov (KS)-like test to determine if genes in a given set are enriched at the top or bottom of a ranked list. **D** Expression of tumor necrosis factor alpha (TNF-α), interleukin-6 (IL-6), granulocyte-macrophage colony stimulating factor (GM-CSF), and leukemia inhibiting factor (LIF) in endothelial cells from ruxolitinib ($n = 6$) vs. vehicle ($n = 5$) treated JAK2-V617F mice as detected by microarray assay. Data represent mean ± SEM. $P$ value was calculated using two-tailed Student's $t$ test. Intracellular FACS analysis of **E** TNF-α+ and **F** GM-CSF+ bone marrow stromal endothelial cells from ruxolitinib ($n = 3$) vs. vehicle ($n = 3/4$) treated empty vector, JAK2-V617F and JAK2-V617F + L902Q mice as detected by microarray assay. Data represent mean ± SEM. $P$ value was calculated using two-tailed Student's $t$ test. Source data is provided in the Source Data file.

**Microarray analysis**

Transcriptome analysis of sorted bone marrow stroma cells as well as sorted bone marrow (EGFP+ or EGFP−CD11b/Gr1+) and splenocytes (EGFP+ or EGFP−Thy1.2+) was performed using a mouse Clariom S Pico

Assay (Thermo Fisher) according to the manufacturer's instructions. RNA quality was assayed using an Agilent 4200 TapeStation (Agilent Technologies, Santa Clara, CA, USA). Analysis was performed by the Transcriptome Analysis Console (TAC, Thermo Fisher).

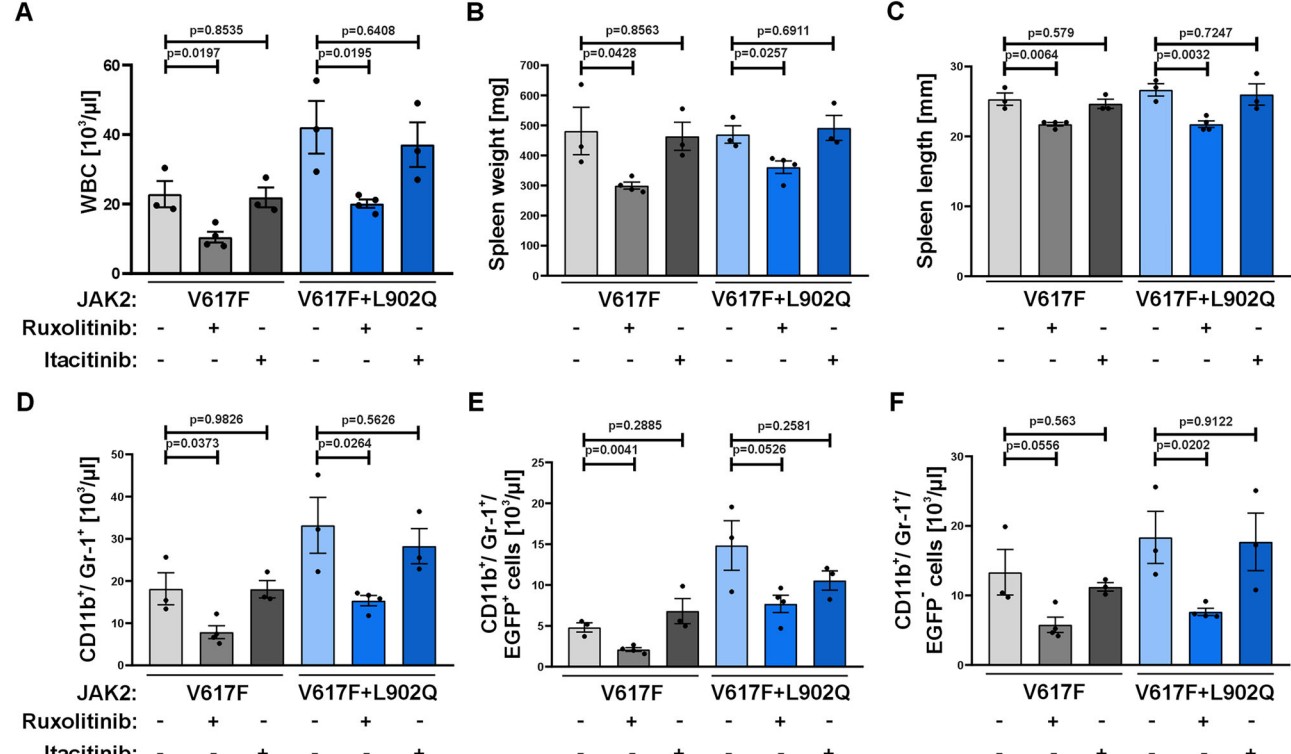

**Fig. 7 | Itacitinib treatment does not attenuate disease phenotype in the ruxolitinib sensitive or resistant mouse model.** While ruxolitinib treatment reduced white blood cell (WBC) count (**A**), spleen weight (**B**) and length (**C**) and total numbers of myeloid cells (**D**) in both the ruxolitinib sensitive JAK2-V617F and ruxolitinib resistant JAK2-V617F + L902Q, itacitinib did not show any effects ($n = 3$ vs. 4 vs. 3 animals per treatment for each JAK2 mutation). Data represent mean ± SEM. $P$ value was calculated using two-tailed Student's $t$ test. In contrast to ruxolitinib, itacitinib neither decreased EGFP+ (**E**) or EGFP– (**F**) CD11b+/Gr-1+ cells. ($n = 3$ vs. 4 vs. 3 animals per treatment for each JAK2 mutation). Data represent mean ± SEM. $P$ value was calculated using two-tailed Student's $t$ test. Source data is provided in the Source Data file.

## Cytokine array

Mouse blood was collected in serum collecting tubes (Microtainer; BD Bioscience) and allowed to clot at room temperature for 30 min before centrifugation ($2200 \times g$, 4 °C, 10 min). Inflammatory cytokine levels in serum were measured using the LEGENDplex™ Mouse Inflammation Panel (13-plex) (BioLegend) according to the manufacturers' instructions. Samples were analyzed in duplicates (25 μl each).

## Statistics and reproducibility

$P$ values were calculated by two-sided Student's $t$-test or one-way ANOVA test using Prism software (GraphPad, La Jolla, CA, USA), unless otherwise specified in the figure legend. Data represent mean ± SEM or Mean ± SD. Sample sizes were estimated in cooperation with the Center for Medical Biometry and Medical Informatics at the University of Freiburg. For in vivo experiments, animals were randomized, and non-blinding was used. All animals with stable engraftment of transplanted bone marrow cells were included into the analyses. All immunoblots are representative of independent experiments.

## Reporting summary

Further information on research design is available in the Nature Portfolio Reporting Summary linked to this article.

## Data availability

The microarray data generated in this study have been deposited at Gene Expression Omnibus under accession codes: GSE272140 and GSE272043. Source data are provided with this paper.

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

## Acknowledgements
The authors would like to thank their funding agencies. The project was supported by the German Research Foundation (DFG) (SFB-1479—Project ID: 441891347 to J.D. and R.Z., RA 3488/1–1 (to M.R). S.P.G. and N.v.B. are supported by DFG (3554/1-3).

## Author contributions
S.P.G. and J.D. designed the study, S.P.G., M.R., T.A.M., T.P., S.M.M.G., S.K.S., K.A.C., H.K., C.E., S.S., D.S., and G.P. performed experiments and analyzed data; I.G.-M. and L.Q.-M. performed and analyzed histological sections; D.B. performed the structural discussion; R.T. performed mass spectrometric analyses of mouse serum; S.R. performed CT scan analyses of mice; D.P. and G.A. performed microarray analyses, R.Z., N.v.B., and A.L.I. provided critical materials, S.P.G., M. R., T.A.M., R.Z., and J.D. wrote the manuscript.

## Funding

## Competing interests
The authors declare no competing interests.

## Additional information

[1]Faculty of Medicine, Clinic for Internal Medicine I, Hematology, Oncology and Stem cell transplantation, University Medical Center Freiburg, Freiburg, Germany. [2]Department of Hematology and Oncology, University Medical Center Schleswig-Holstein, and University Cancer Center Schleswig-Holstein, Lübeck, Germany. [3]Laboratory of Regenerative Immunotherapy, Department of Cell Growth and Differentiation, Center for iPS cell Research, Kyoto University, Kyoto, Japan. [4]Faculty of Biology, University of Freiburg, Freiburg, Germany. [5]Department I of Internal Medicine, Center for Integrated Oncology, Aachen-Bonn-Cologne-Duesseldorf, University of Cologne, Cologne, Cologne, Germany. [6]Department of Pathology and Neuropathology, University Hospital Tübingen & Comprehensive Cancer Center Tübingen, Tübingen, Germany. [7]Cluster of Excellence iFIT (EXC 2180) "Image-Guided and Functionally Instructed Tumor Therapies", Eberhard-Karls University of Tübingen, Tübingen, Germany. [8]Institute of Physiology, Albert-Ludwigs-Universität Freiburg, Freiburg, Germany. [9]Institute of Pharmaceutical Sciences, Albert-Ludwigs-Universität Freiburg, Freiburg, Germany. [10]Faculty of Medicine, Department of Nuclear Medicine, University Medical Center Freiburg, Freiburg, Germany. [11]German Cancer Consortium (DKTK) and German Cancer Research Center (DKFZ), Heidelberg, Germany. [12]Institute of Medical Bioinformatics and Systems Medicine, Medical Center, Faculty of Medicine, University of Freiburg, Freiburg, Germany. [13]These authors contributed equally: Sivahari Prasad Gorantla, Michael Rassner. ✉e-mail: justus.duyster@uniklinik-freiburg.de

