## [Transparent Peer Review file · Nature Communications]

Efficacy of JAK1/2 inhibition in murine myeloproliferative neoplasms is not mediated by targeting oncogenic signaling

Corresponding Author: Professor Justus Duyster

Version 0:

Reviewer comments:

Reviewer #1

(Remarks to the Author)

In their manuscript "Efficacy of JAK1/2 inhibition in murine myeloproliferative neoplasms is mediated mainly by targeting pro-inflammatory cytokine signaling not oncogenic signaling" The authors very rightly point towards the therapeutic problem in the field – being that the ruxolitinib, a commonly used JAK1/JAK2 inhibitor for primary myelofibrosis (PMF), although reduces spleen size and alleviates leukocytosis, it does not improve fibrosis, nor does it affect the allelic burden. The authors suggest that the reduction in spleen size and leukocyte count seen with ruxolitinib is independent of the oncogene. The authors thus attempt to identify the mechanism of action of ruxolitinib.

The authors develop two mouse models which are resistant to ruxolitinib at varying degree, though this is shown only with respect to levels of STAT5 signaling, which means that the actual impact on oncogenic activity (ie, aberrant stem cell replicative function, serial CFU activity, signaling via STAT3 and other pathways that compensate for JAK inhibition in MPN, etc) is not measured. This lack of functional validation of the mutations undermines the key claim in the title, as the provided mouse model data themselves are not otherwise able to distinguish between inflammatory versus oncogenic activity. The reader is left to assume that the mutations made by the authors actually are sufficient to block ruxolitinib activity all round, as well as to compensate for blockade of signaling via endogenous JAK2. As the mutant is not introduced into WT JAK2, a key mechanistic control is thus also missing.

The authors show that the regular mouse model with the commonly found JAK2-V617F mutation with ruxolitinib treatment shows similar phenotype as the humans – where treatment reduces spleen size, and reduces leukocyte count, with no effect on fibrosis (Figure 3). As the phenotype is no different in the two mutant models, it is again unclear whether the results are due to impact on inflammation versus oncogenesis because the mouse models are not functionally validated as above.

The authors' assertion that inflammation is the critical driver of MPN in this setting is made indirectly, ie by measuring levels of inflammatory cytokines. However, these data are associative and not causative, and no new mechanistic insight into how the mutations and/or ruxolitinib modulate inflammation is provided.

The authors allude that the potential mechanism could be through stromal cells, however, to make the claim, validation experiments are needed, as transcript expression does not always translate to active biology.

Hence, altogether while the overall hypothesis posed by the authors is intriguing, insufficient experimental evidence to provide a direct demonstration of their proposed mechanism and to exclude alternative hypotheses has not been provided.

Some other comments to improve the manuscript are as follows:

- Figure 3, 4, 5: The authors are missing WT/mock controls in all their experiments. Kindly provide control data as well.
- Figure 1A, D S2B: Kindly add loading controls and provide quantification of the blots.
- Figure 1C, 2A, 2B, 2E, S3A-D: Statistical analysis is missing on these figures. Kindly add them.
- Figure 2F, 3G, 5E: The image is missing the scale bar. Kindly add for readers' reference.
- Figure 2F: The authors either provide quantification or show with arrows to complement their statement – “marked infiltration by hematopoietic cells with a left shifted granulopoiesis, erythropoiesis, and moderately increased megakaryopoiesis”
- Figure 3E-F: The authors show in fig 2E that both their PMF mouse models have increased myeloid cells and reduced B

cells. In figure 3E-F, the authors show that upon ruxolitinib treatment, the myeloid cell count reduces, does the B-cell count increase to go back to normal?

- Figure 4A-E: Kindly specify is the serum analysis in bone marrow or peripheral blood serum.
- Figure 4F-J: The authors should include the unit of measurement in the Y-axis.
- Figure 5E: The authors provide quantification for the V617F mouse model in 5F, but a representative image for visual comparison is missing in 5E.
- Figure 5G: The authors should clarify if this fig is representative of all mouse models used by them.
- Figure 6A: Kindly provide clarification if mesenchymal stem cells or endothelial cells were used to generate the enrichment analysis plot.

Reviewer #2

(Remarks to the Author)

This is a nicely designed and performed study which aims to look at the relative contribution of mutant and wild-type cells to the efficacy of Ruxolitinib in a JAK2V617F mutant model of MPN. The work is well done and presented, but some important questions remain to be addressed.

1. The work here with V617F + rux resistance mutations suggests, as the authors surmise, that JAK2 inhibition has non effect on the mutant cells, and only impacts the WT cells. This is an interesting, even tantalizing idea, but there are several things to consider. One, in the second mutant model, the impact of Ruxolitinib in reducing spleen weight is less significant in the resistance model compared to the V617F mutant alone model. How is this reconciled with the idea that there is no effect on the mutant cells? Have the authors measured JAK2 and STAT5 phosphorylation in vivo to show degree, or lack thereof, of target inhibition?
2. Second, previous studies have shown that Ruxolitinib can reduce malignant cytokine production from both mutant and WT cells (shown both in the MPL mutant model and in the V617F constitutive KI model). How do the authors reconcile their data with this previous work? Could the inability of Ruxolitinib to reduce cytokine production from mutant cells in their model reflect the very high levels of V617F + second-site mutant expression in an ectopic BMT model? Might the lower levels of mutant expression of V617F be inhibited by Ruxolitinib when the mutant is expressed from the endogenous locus?
3. Likewise, it would be important to show that mRNA of key cytokines which do/don't change with Ruxolitinib treatment in vivo in the murine model have similar effects in clinical isolates from patients treated with Ruxolitinib. E.g. does Ruxolitinib not reduce these key cytokines expression from human samples on therapy->ideally from patients with high JAK2V617F VAF where one can be sure that one is largely querying mutant cells.
4. If the authors transplant V617F mutant cells with JAK2V617-wildtype cells with a resistance mutation->does Ruxolitinib still work? E.g. is the key cytokine signal from hematopoietic cells (versus stromal cells which cannot be tested).

Version 1:

Reviewer comments:

Reviewer #1

(Remarks to the Author)

Overall the authors have done an excellent job revising the manuscript to make a clearer demonstration that JAK2 inhibition does not directly target the mutant allele fraction, using a combination of mutants that are resistant to JAK2 inhibition. These revised and extended data have bolstered the authors' main conclusion.

However, the extent to which blockade of stromal inflammation is the relevant player is still less clearly elucidated and the mechanism proposed by the authors while interesting is not conclusive and remains a largely associative component of the study. We recognize that a comprehensive dissection of the mechanism is beyond the scope of what can be included in a single paper.

We would recommend that the authors retitle their manuscript to reflect their main and concrete conclusion - that JAK inhibition does not target oncogenic signaling. The data do not yet support the conclusion that inflammation is thus the target of JAK inhibition beyond association.

Reviewer #2

(Remarks to the Author)

Very nicely revised with key experiment done with resistance mutation in both mutant and WT cells. No further comments.

Reviewer comments:

Reviewer #1 (Remarks to the Author):

The authors develop two mouse models which are resistant to ruxolitinib at varying degree, though this is shown only with respect to levels of STAT5 signaling, which means that the actual impact on oncogenic activity (ie, aberrant stem cell replicative function, serial CFU activity, signaling via STAT3 and other pathways that compensate for JAK inhibition in MPN, etc) is not measured. This lack of functional validation of the mutations undermines the key claim in the title, as the provided mouse model data themselves are not otherwise able to distinguish between inflammatory versus oncogenic activity. The reader is left to assume that the mutations made by the authors actually are sufficient to block ruxolitinib activity all round, as well as to compensate for blockade of signaling via endogenous JAK2. As the mutant is not introduced into WT JAK2, a key mechanistic control is thus also missing.

Response: We have now functionally validated the resistance mutations by measuring CFU activity and signaling via STAT3 and other pathways in JAK2-V617F cells that carry ruxolitinib resistance mutations (Supplemental Figure 4, 6) as suggested by the reviewer. Both JAK2-V617F+L902Q and JAK2-V617F+L983F displayed similar numbers of BFU-E, CFU-E, and CFU-GM compared to JAK2-V617F (Supplementary Fig. 6A-C). Moreover, while STAT3 phosphorylation was completely blocked by ruxolitinib in ruxolitinib-sensitive JAK2-V617F cells, it was not significantly reduced in JAK2-V617F cells that additionally carry the ruxolitinib resistance mutations despite ruxolitinib exposure (Supplemental Figure 4). Additionally, as suggested by the reviewer, we introduced the ruxolitinib resistance mutations into WT JAK2 and expressed it in Ba/F3 cells. When treated with ruxolitinib, ruxolitinib-resistant cells did not show reduction of proliferation compared to ruxolitinib-sensitive cells and reduction of STAT5, STAT3, AKT and ERK activation upon exposure to 2 μ m ruxolitinib was reduced compared to WT JAK2 (Supplemental Figure 17). This data show that the mutations are conferring resistance to ruxolitinib in the context of WT JAK2 and JAK2-V617F.

The authors show that the regular mouse model with the commonly found JAK2-V617F mutation with ruxolitinib treatment shows similar phenotype as the humans – where treatment reduces spleen size, and reduces leukocyte count, with no effect on fibrosis (Figure 3). As the phenotype is no different in the two mutant models, it is again unclear whether the results are due to impact on inflammation versus oncogenesis because the mouse models are not functionally validated as above.

Response: We agree with the reviewer and now discuss that the effects by ruxolitinib are most likely due to the reduction of inflammation, because we can exclude that the oncogenic

downstream signaling is still fully active due to the resistance mutation protecting it from the effect of ruxolitinib.

Indeed, it was shown within the last years by us and others that ruxolitinib works anti-inflammatory in humans, e.g. by reducing cytokine signaling (Spoerl et al., 2014 *Blood*; Zeiser et al., 2015 *Leukemia*, 2020 *NEJM*).

The authors' assertion that inflammation is the critical driver of MPN in this setting is made indirectly, ie by measuring levels of inflammatory cytokines. However, these data are associative and not causative, and no new mechanistic insight into how the mutations and/or ruxolitinib modulate inflammation is provided.

Response: We thank the reviewer for this valuable comment. Indeed, there are several reports within the last years that have investigated the role of inflammation, in particular cytokines, in myeloproliferative neoplasms (MPN): For example, it was shown that anti-IL-1 β antibody treatment reduced myelofibrosis and osteosclerosis in JAK2-V617F mice (Rai et al., *Nat Commun* 2022). In a phase Ib clinical trial of a TGF β 1/3 trap in advanced MF, patients showed spleen and symptom benefits in a few patients (Mascarenhas et al., *Clin Cancer Res* 2023). Other prior work has identified the ERK/Sp1/TGF β 1 axis in CALRdel52 MPNs as a mechanism of immunosuppression (Schmidt et al., *Cancer Res* 2024). Another study showed that TNF α facilitates clonal expansion of JAK2-V617F positive cells in MPN (Fleischman et al., *Blood* 2011). TNF- α levels correlated with the JAK2-V617F allele burden and exposure of JAK2-V617F-positive cells to a JAK inhibitor caused reduced TNF- α transcription. Using TNF- α deficient mice, the authors could demonstrate that TNF- α was required for the development of the MPN-like disease.

These studies indicated that elevated cytokines are not only associative but also causative for MPN. We published an overview article addressing this interesting question (Braun and Zeiser: Immunotherapy in Myeloproliferative Diseases. *Cells* 2020). We have now included this important point in the discussion.

The authors allude that the potential mechanism could be through stromal cells, however, to make the claim, validation experiments are needed, as transcript expression does not always translate to active biology.

Response: We agree with the reviewer that transcript expression cannot always be directly translated to active biology. Therefore, we repeated the *in vivo* experiments and transplanted BM transduced with an empty vector, JAK2-V617F +/- ruxolitinib-resistance mutations, and treated these mice with vehicle vs. ruxolitinib. We then performed flow cytometry for intracellular cytokines

on bone marrow stromal cells and saw a significant reduction of cytokine levels in stromal cells from both ruxolitinib-sensitive and –resistant mice (Figure 6D, E Supplementary Figure 15). This data show that the resistance mutation does not prevent the reduction of cytokine production in stromal cells.

- *Figure 3, 4, 5: The authors are missing WT/mock controls in all their experiments. Kindly provide control data as well.*

Response: We have now provided control data for these figures. We performed additional mouse experiments including mice receiving BM transduced with an empty vector (Supplementary Figure 12). These mice do not develop any disease in terms of WBC, RBC, splenomegaly, etc. Even in these mice - that do not have the JAK2-V617F oncogene - ruxolitinib reduced the spleen size (Supplementary Figure 12A+B). This data indicate some level of basic inflammation in these mice under steady-state conditions (Gasteiger et al., Science 2016) which is reduced by ruxolitinib treatment. Moreover, levels of serum cytokines – which were overall lower than in JAK2-V617 mice – were not affected by ruxolitinib.

- *Figure 1A, D S2B: Kindly add loading controls and provide quantification of the blots.*

Response: We have added loading controls and provide quantification of the blots (Figure 1, Supplemental Figures 2, 3, 4, and 5).

- *Figure 1C, 2A, 2B, 2E, S3A-D: Statistical analysis is missing on these figures. Kindly add them.*

Response: We have added statistical analysis on figures 1C, 2A, 2B, 2E, S3A-D.

- *Figure 2F, 3G, 5E: The image is missing the scale bar. Kindly add for readers' reference.*

Response: We have added the scale bar.

- *Figure 2F: The authors either provide quantification or show with arrows to complement their statement – “marked infiltration by hematopoietic cells with a left shifted granulopoiesis, erythropoiesis, and moderately increased megakaryopoiesis”*

Response: We have included arrows that point towards marked infiltration by hematopoietic cells with a left shifted granulopoiesis as G, erythropoiesis as E, and moderately increased megakaryopoiesis as M (Figure 2). We provide a high-magnification-version for better visibility in the Supplementary Figure 8.

- *Figure 3E-F: The authors show in fig 2E that both their PMF mouse models have increased myeloid cells and reduced B cells. In figure 3E-F, the authors show that upon ruxolitinib treatment, the myeloid cell count reduces, does the B-cell count increase to go back to normal?*

Response: We have provided the requested B-cell counts. Similarly to myeloid cells, B cells counts were reduced in the ruxolitinib treated animals (Supplementary Figure 14).

- *Figure 4A-E: Kindly specify is the serum analysis in bone marrow or peripheral blood serum.*

Response: We have specified that the serum analysis was peripheral blood serum.

- *Figure 4F-J: The authors should include the unit of measurement in the Y-axis.*

Response: We have included the unit of measurement in the Y-axis.

- *Figure 5E: The authors provide quantification for the V617F mouse model in 5F, but a representative image for visual comparison is missing in 5E.*

Response: We provide a new representative image for visual comparison in Figure 5E.

- *Figure 5G: The authors should clarify if this fig is representative of all mouse models used by them.*

Response: We have included a statement that Figure 5G is representative of all mouse models used.

- *Figure 6A: Kindly provide clarification if mesenchymal stem cells or endothelial cells were used to generate the enrichment analysis plot.*

Response: We have used bulk BM stromal cells with the following markers to generate the enrichment analysis plot: negative for CD45/TER-119/CD3/Gr-1 (lineage), positive for CD31, Sca-1, CD166 or CD140. We have included a gating strategy of endothelial cells in Supplementary Figure 15.

Reviewer #2 (Remarks to the Author):

1. The work here with V617F + rux resistance mutations suggests, as the authors surmise, that JAK2 inhibition has non effect on the mutant cells, and only impacts the WT cells. This is an interesting, even tantalizing idea, but there are several things to consider. One, in the second mutant model, the impact of Ruxolitinib in reducing spleen weight is less significant in the resistance model compared to the V617F mutant alone model. How is this reconciled with the idea that there is no effect on the mutant cells? Have the authors measured JAK2 and STAT5 phosphorylation in vivo to show degree, or lack thereof, of target inhibition?

Response: The level of spleen weight reduction seemed less pronounced in the second resistance model as compared to the JAK2V617F model alone (Figure 5A). Mice transplanted with JAK2V617F+L983F⁺ BM depicted in the figures had a *per se* slightly higher spleen weight. However, percentagewise the degree of spleen weight reduction by ruxolitinib between JAK2V617F alone vs. JAK2V617F+L983F was similar (Figure 3, 5).

We have now measured STAT5 phosphorylation *in vivo*. For this purpose, we have transplanted EGFP⁺ JAK2-V617F cells (vs. JAK2-V617F+L902Q vs. empty vector) together with EYFP⁺ empty vector cells. The results showed inhibition of STAT5 phosphorylation in the EYFP⁺ population, i.e. in cells that do not have the oncogene, rather than EGFP⁺ population in JAK2-V617F and JAK2-V617F+L902Q animals. These results are shown in Supplemental Figure 11.

2. Second, previous studies have shown that Ruxolitinib can reduce malignant cytokine production from both mutant and WT cells (shown both in the MPL mutant model and in the V617F constitutive KI model). How do the authors reconcile their data with this previous work? Could the inability of Ruxolitinib to reduce cytokine production from mutant cells in their model reflect the very high levels of V617F + second-site mutant expression in an ectopic BMT model? Might the lower levels

of mutant expression of V617F be inhibited by Ruxolitinib when the mutant is expressed from the endogenous locus?

Response: We agree that ruxolitinib can reduce malignant cytokine production from both mutant and WT cells as shown for instance by Kleppe et al. (2015). In this study, the authors have shown that certain cytokines are reduced in different cell compartments (mutant vs. nonmutant hematopoietic cells) upon ruxolitinib treatment. Similarly, TNF- α transcripts were reduced in JAK2-V617F+ BM cells from ruxolitinib-treated mice, whereas other cytokines and chemokines such as INF- γ , IL-10 or CCL-3 were lower in myeloid cells of non-malignant origin. Thus, our novel finding is that the relative contribution of cytokine production from WT cells for the disease phenotype is more relevant. This was only implied by Kleppe et al.'s study where pan-hematopoietic (mutant plus nonmutant) but not MPN-specific *Stat3* deletion alleviated disease activity including cytokines production in MPL^{W515L} MPN mice. Additionally, the authors used a *Jak2*^{V617F};*Vav-Cre* knockin model in contrast to us and the extent by which ruxolitinib reduced cytokine expression from Jak2V617F mutant vs. nonmutant CD45⁺ cells was not analyzed (s. Kleppe et al. (2015), Suppl. Figure 15).

To further delineate the relative contribution, we have now performed a new transplant experiment: We first cloned the JAK2-L902Q and introduced this resistance mutation in the non-oncogenic JAK2 alone into hematopoietic BM cells. Mice were then transplanted with JAK2-L902Q⁺ BM cells alongside oncogenic ruxolitinib sensitive JAK2-V617F⁺ BM cells (JAK2-V617F/JAK2-L902Q chimera) and treated with ruxolitinib. Interestingly, mice harboring the ruxolitinib resistance mutation in their non-malignant hematopoietic cells displayed reduced response to ruxolitinib compared to those with a mock control (Supplemental Figure 18). These experiments further support our notion, that susceptibility towards JAK inhibition in this mouse model depends more on the nonmalignant hematopoietic cells than the malignant hematopoietic cells. In addition, our other experiments suggest that also nonmalignant stroma cells do contribute to the ruxolitinib response.

3. Likewise, it would be important to show that mRNA of key cytokines which do/don't change with Ruxolitinib treatment in vivo in the murine model have similar effects in clinical isolates from patients treated with Ruxolitinib. E.g. does Ruxolitinib not reduce these key cytokines expression from human samples on therapy->ideally from patients with high JAK2V617F VAF where one can be sure that one is largely querying mutant cells.

Response: We are thankful for the reviewer suggestion. We now include a discussion on the literature showing that ruxolitinib reduced many but not all cytokines to normal levels (e.g. Verstovsek et al., *N Engl J Med* 2010; Quintás-Cardama et al., *Blood* 2010). This can be explained by the fact that cytokines are a product from mutant hematopoietic cells and nonmutant

hematopoietic cells and stromal cells (Figure 4, 6, Suppl. Figure 10, 16). Thus, even though mRNA levels of some cytokines do not change in one cell type, they might do so in other cells.

In addition, we now discuss a study by Fisher et al. demonstrating that MF patients exhibit significant upregulation of inflammatory cytokines including VEGF, IL-10, TNF, IL-16 and IL-6, and that ruxolitinib treatment did not reduce all the cytokine levels equally.

4. If the authors transplant V617F mutant cells with JAK2V617-wildtype cells with a resistance mutation->does Ruxolitinib still work? E.g. is the key cytokine signal from hematopoietic cells (versus stromal cells which cannot be tested).

Response: We thank the reviewer for this interesting suggestion. We believe the reviewer asked for a combinatory transplantation of JAK2-V617F together with JAK2-L902Q. We have transplanted JAK2-V617F cells together with JAK2-wildtype cells with a resistance mutation (vs. empty vector) and tested for ruxolitinib responses (please s. comment 2). Mice transplanted with JAK2-V617F cells together with ruxolitinib resistant JAK2-L902Q bone marrow (JAK2-V617F/JAK2-L902Q chimera) developed an MPN-like disease. However, ruxolitinib treatment did not significantly reduce the Hgb or HCT compared to JAK2-V617F/empty chimera mice. In addition, statistically significant inhibition of STAT5 phosphorylation was not observed in JAK2-V617F/JAK2-L902Q PB myeloid cells compared to JAK2-V617F/empty mice. This new data indicate that ruxolitinib in MPN is predominantly suppressing the WT-JAK2 mediated signaling in non-malignant hematopoietic cells rather than malignant cells. These results are shown in Supplementary Figure 18.

In summary, we hope to have addressed the reviewer comments sufficiently and look forward to your response.

Sincerely,

Justus Duyster, M.D.

Department of Hematology and Oncology

University Medical Center Freiburg

Literature

Baldauf, CK et al. Anti-IL-6 cytokine treatment has no impact on elevated hematocrit and splenomegaly in a polycythemia vera mouse model. *Blood Adv.* bloodadvances.2021004379 (2021)
doi:10.1182/bloodadvances.2021004379

Fisher DAC, et al. Cytokine production in myelofibrosis exhibits differential responsiveness to JAK-STAT, MAP kinase, and NFκB signaling. *Leukemia* 33, 1978-1995 (2019).

Fleischman, AG et al. TNFα facilitates clonal expansion of JAK2V617F positive cells in myeloproliferative neoplasms. *Blood* 118; 6392-8 (2011)

Gasteiger, G et al. Tissue residency of innate lymphoid cells in lymphoid and non-lymphoid organs. *Science* 350, 981-985 (2016)

Hasselbalch, HC, Bjorn, ME. Ruxolitinib versus standard therapy for the treatment of polycythemia vera. *N Engl J Med* 372, 1670 (2015)

Kleppe, M et al. JAK-STAT pathway activation in malignant and nonmalignant cells contributes to MPN pathogenesis and therapeutic response. *Cancer Discov* 5, 316-31 (2015)

Mascarenhas, J et al. A Phase Ib Trial of AVID200, a TGFβ 1/3 Trap, in Patients with Myelofibrosis. *Clin. Cancer Res. Off. J. Am. Assoc. Cancer Res.* 29, 3622–3632 (2023)

Quintás-Cardama, A et al. Preclinical characterization of the selective JAK1/2 inhibitor INCB018424: therapeutic implications for the treatment of myeloproliferative neoplasms. *Blood* 115:3109-3117 (2010)

Rai, S. et al. Inhibition of interleukin-1β reduces myelofibrosis and osteosclerosis in mice with JAK2-V617F driven myeloproliferative neoplasm. *Nat. Commun.* 13, 5346 (2022)

Schmidt, D et al. Oncogenic Calreticulin Induces Immune Escape by Stimulating TGFβ Expression and Regulatory T-cell Expansion in the Bone Marrow Microenvironment. *Cancer Res.* 16, 2985-3003 (2024)

Spoerl, S et al. Activity of therapeutic JAK 1/2 blockade in graft-versus-host disease. *Blood* 12, 3832-42 (2014)

Verstovsek, S et al. Safety and efficacy of INCB018424, a JAK1 and JAK2 inhibitor, in myelofibrosis. *N Engl J Med.* 16, 1117-27 (2010)

Zeiser, R et al. Ruxolitinib in corticosteroid-refractory graft-versus-host disease after allogeneic stem cell transplantation: a multicenter survey. *Leukemia* 29: 2062-8 (2015)

Zeiser R, et al. Ruxolitinib for Glucocorticoid-Refractory Acute Graft-versus-Host Disease. *NEJM* 383, 1800-1810 (2020)

Reviewer comments:

Reviewer #1 (Remarks to the Author):

Overall the authors have done an excellent job revising the manuscript to make a clearer demonstration that JAK2 inhibition does not directly target the mutant allele fraction, using a combination of mutants that are resistant to JAK2 inhibition. These revised and extended data have bolstered the authors' main conclusion.

However, the extent to which blockade of stromal inflammation is the relevant player is still less clearly elucidated and the mechanism proposed by the authors while interesting is not conclusive and remains a largely associative component of the study. We recognize that a comprehensive dissection of the mechanism is beyond the scope of what can be included in a single paper.

We would recommend that the authors retitle their manuscript to reflect their main and concrete conclusion - that JAK inhibition does not target oncogenic signaling. The data do not yet support the conclusion that inflammation is thus the target of JAK inhibition beyond association.

Response: We thank the reviewer for his/her comments. We have retitled our manuscript that JAK inhibition does not target oncogenic signaling and have, thus, removed the effects on inflammation.

Reviewer #2 (Remarks to the Author):

Very nicely revised with key experiment done with resistance mutation in both mutant and WT cells. No further comments.

Response: We thank the reviewer for his/her constructive comments during the revisions and the very kind remark.

In summary, we hope to have addressed the reviewer comments sufficiently and look forward to your response.

Sincerely,

Justus Duyster, M.D.

Department of Hematology and Oncology

University Medical Center Freiburg